# $\mathbb{D}^2$ PRUNING: MESSAGE PASSING FOR BALANCING DIVERSITY & DIFFICULTY IN DATA PRUNING

**Adyasha Maharana, Prateek Yadav & Mohit Bansal**
Department of Computer Science
University of North Carolina at Chapel Hill
{adyasha,praty,mbansal}@cs.unc.edu

## ABSTRACT

In recent years, data quality has emerged as an important factor for training massive models. Analytical theories suggest that higher-quality data can lead to lower test errors in models trained on a fixed data budget. Moreover, a model can be trained on a lower compute budget without compromising performance if a dataset can be stripped of its redundancies. Coreset selection (or data pruning) seeks to select a subset of the training data so as to maximize the performance of models trained on this subset, also referred to as coreset. There are two dominant approaches: (1) geometry-based data selection for maximizing *data diversity* in the coreset, and (2) functions that assign *difficulty scores* to samples based on training dynamics. Optimizing for data diversity leads to a coreset that is biased towards easier samples, whereas, selection by difficulty ranking omits easy samples that are necessary for the training of deep learning models. This demonstrates that data diversity and importance scores are two complementary factors that need to be jointly considered during coreset selection. In this work, we represent a dataset as an undirected graph and propose a novel pruning algorithm, $\mathbb{D}^2$ PRUNING, that uses message passing over this dataset graph for coreset selection. $\mathbb{D}^2$ PRUNING updates the difficulty scores of each example by incorporating the difficulty of its neighboring examples in the dataset graph. Then, these updated difficulty scores direct a graph-based sampling method to select a coreset that encapsulates both diverse and difficult regions of the dataset space. We evaluate supervised and self-supervised versions of our method on various vision and NLP datasets. Results show that $\mathbb{D}^2$ PRUNING improves coreset selection over previous state-of-the-art methods at low-to-medium pruning rates. Additionally, we find that using $\mathbb{D}^2$ PRUNING for filtering large multimodal datasets leads to increased diversity in the dataset and improved generalization of pretrained models. Our work shows that $\mathbb{D}^2$ PRUNING is a versatile framework for understanding and processing datasets.[1]

## 1 INTRODUCTION

Deep learning models are evolving into massive architectures with trillions of learnable parameters requiring enormous training datasets for optimal performance. Empirical experiments demonstrate that the test error in such models falls off as a power law with model size as well as training dataset size (Kaplan et al., 2020). Recently, Sorscher et al. (2022) developed an analytical theory that shows that the power law association of test error with data size can be demoted to exponential scaling if one has access to a high-quality *data pruning* metric for careful data selection. This has the implication that for a fixed data budget, high-quality training data can yield lower test loss in deep learning models. *Coreset selection* [2] (Mirzasoleiman et al., 2020; Guo et al., 2022) is a similar line of work that aims to select a subset (coreset) of the most informative samples $\mathcal{S}$ from a large training dataset $\mathcal{T}$ without significantly compromising the performance of the model. Existing coreset selection methods (Toneva et al., 2018; Killamsetty et al., 2021a;b; Yang et al., 2022; Sorscher et al., 2022) demonstrate promising performance on many vision datasets for one-shot coreset selection. However, significant

---

[1]Our code is available at https://github.com/adymaharana/d2pruning
[2]We use the terms coreset selection and data pruning interchangeably throughout the paper.

progress remains to be made on the selection of better coresets, especially using self-supervised approaches. Moreover, there is a lack of systematic evaluation of these methods on NLP datasets.

Real-world data distributions comprise high-density as well as low-density regions. Yu et al. (2020); Chan et al. (2022) claim that maximizing the variance of intra-class features results in robust representations. To this end, geometry-based coreset selection methods (Sener & Savarese, 2018; Chen et al., 2010) operate under the assumption that samples located close to each other provide redundant information, and try to remove those data points by selecting the samples most distant from $k$-means cluster centers (Sorscher et al., 2022) or at a median distance from the class center (Xia et al., 2023), in order to maximize diversity in the coreset. On the other hand, uncertainty-based methods (Coleman et al., 2019) and error or loss-based methods (Toneva et al., 2018; Paul et al., 2021) propose a score-based function to estimate the difficulty of each sample in the training dataset from the model's training dynamics and retain the most difficult samples. However, the distribution of difficulty scores for the original data is highly skewed and contain way more low-difficulty (or easy) samples (Swayamdipta et al., 2020), as we show in Figure 2(a). As low-difficulty samples predominantly arise in densely populated regions (Sorscher et al., 2022), incorporating some of these well-connected, low-difficulty samples into the coreset guarantees adequate representation of these dense areas within the coreset (Zheng et al., 2022). At the same time, selecting high-difficulty samples with higher connectivity increases the information content of the (Kim & Shin, 2022). Evidently, *example difficulty* and *data diversity* are two crucial factors for selecting effective coresets, yet, there has been little work towards combining them into a unifying framework for coreset selection.

To unify these two factors, we propose the $\mathbb{D}^2$ PRUNING method, where we represent the dataset $\mathcal{S}$ as an undirected graph $\mathcal{G}$ and design a message-passing algorithm that unifies the difficulty scores and the underlying spatial distribution of the dataset to select a coreset with balanced difficulty and diversity. $\mathbb{D}^2$ PRUNING consists of three simple steps: (1) **Graph Initialization:** First, we create a *graph*, $\mathcal{G}$, where each node is an example from the dataset $\mathcal{S}$ and is connected to its $k$-closest neighbors based on a notion of distance in the embedding space (see Fig. 1(A)). Each node has a feature value that represents the *difficulty score* of the example. This graph can be used to understand the connectivity of each sample with respect to the rest of the dataset (Ebert et al., 2012). (2) **Forward Message Passing:** Next, we perform message passing (Gasteiger et al., 2020; Yadav et al., 2019) over the dataset graph to update the difficulty scores of all examples by taking into account the distance and difficulty of its neighboring examples in the graph (see Fig. 1(B)). Specifically, each node collects a message from all of its neighbors (where the message is their difficulty scores scaled by their distance) and uses these messages to update its own difficulty score. (3) **Coreset Selection & Reverse Message Passing:** Finally, we use these updated scores to iteratively select a balanced subset of samples from high-density low-difficulty regions and low-density high-difficulty regions. At each step of selection, the neighbors of the selected sample are down-weighted via reverse message-passing to promote diversity in the coreset (see Fig. 1(C)). Our design ensures that highly connected nodes of low difficulty are on equal footing with sparsely connected nodes of high difficulty during selection.

We refer to this diversity-difficulty ($\mathbb{D}^2$) approach of coreset selection using message-passing as $\mathbb{D}^2$ PRUNING and evaluate this pruning method on multiple image classification and natural language processing (NLP) datasets. We find that $\mathbb{D}^2$ PRUNING outperforms state-of-art methods for coreset selection at low-to-medium pruning rates. Our analysis shows that $\mathbb{D}^2$ PRUNING selects a coreset with a higher distribution of difficult samples for low pruning rates and with equitable distribution over easy and difficult samples for medium-to-high pruning rates. Further, we adapt $\mathbb{D}^2$ PRUNING for self-supervised and unsupervised data selection approaches and show improvements over existing methods for self-supervised coreset selection and data filtering respectively. Importantly, the message-passing framework for coreset selection opens up possibilities for exploring different message schemes, possibly incorporating factors other than data diversity and difficulty, in an easy plug-and-play framework. In summary, our contributions are:

- We propose $\mathbb{D}^2$ PRUNING, a one-shot coreset selection algorithm that represents datasets as undirected graphs and uses message-passing to combine the influence of two important factors, *example difficulty* and *data diversity*, for data selection.

- We evaluate our method on several image classification, NLP benchmarks and show state-of-the-art results for low-to-medium pruning rates for supervised & self-supervised approaches.

- We show that $\mathbb{D}^2$ PRUNING selects diverse data pools when filtering massive multimodal datasets, which improves the generalization of pretrained multimodal models.

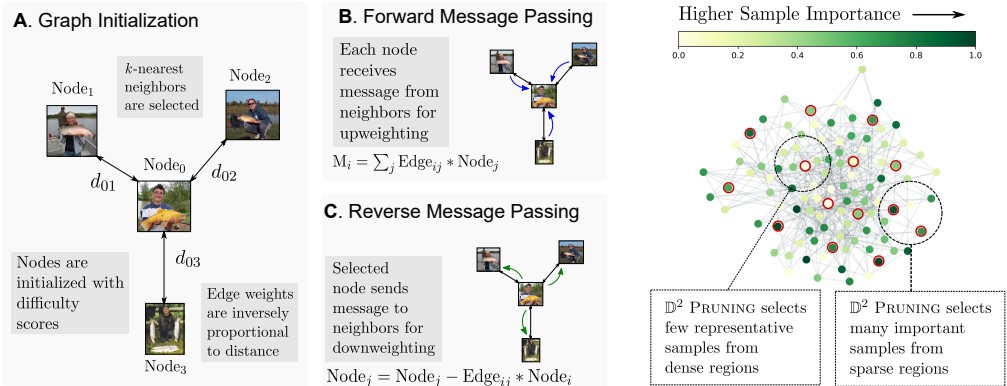

Figure 1: Overview of $\mathbb{D}^2$ PRUNING. (left) Our proposed algorithm contains three steps: (a) Initialization of graph $\mathcal{G}$ using difficulty scores and edge weights based on feature distance, (b) message passing between connected nodes to propagate difficulty scores of neighboring samples, and (c) data selection and reverse message passing to avoid sampling from the same neighborhood. (right) $\mathbb{D}^2$ PRUNING selects a balanced subset of samples (red) from sparse and dense regions.

## 2    PRELIMINARIES

In this section, we describe one-shot coreset selection and discuss the motivation behind our work.

### 2.1    ONE-SHOT CORESET SELECTION

Consider a training dataset $S$ containing $N$ examples $\{(x_i, y_i)\}_{i=1}^{N}$ drawn i.i.d. from an underlying distribution $P$. One-shot coreset selection refers to the selection of a subset $S'$ of the data at a given pruning rate $\alpha$ such that the loss of the model $\theta$ trained on $S'$ using loss function $L$ is minimized on an evaluation set drawn from $P$. This results in the optimization problem as follows:

$$\min_{S' \subset S:\, \frac{|S'|}{|S|} \leq (1-\alpha)} E_{x,y \sim P}[L(x, y; \theta^*(S'))] \tag{1}$$

### 2.2    DESIDERATA OF CORESET

Coresets are representative subsets of larger datasets and aim to preserve the performance achieved by training on the full dataset. Prior works on understanding training dynamics point towards two important factors for ensuring the same i.e. example difficulty and data diversity.

**Example difficulty.** Multiple works have sought to define example difficulty in order to understand how deep neural networks process data. Statistical metrics like consistency score (Jiang et al., 2021) measure the probability of predicting the correct label of an instance when it is left out of the training dataset. Sorscher et al. (2022) provide theoretical justification for retaining the hardest examples when pruning large datasets for a perceptron learning setting. Swayamdipta et al. (2020) show that examples that have a high degree of variance in the model's predictions during training have the largest impact on the model's overall performance. Accordingly, coreset selection methods based on difficulty score functions prioritize the selection of difficult examples for coresets (Guo et al., 2022). However, it has been shown that deep learning models learn easy data and simple functions earlier in training (Jiang et al., 2021; Toneva et al., 2018; Baldock et al., 2021) and easy examples ease the optimization of deep learning networks in the high-dimensional data manifold. Moreover, Zheng et al. (2022) demonstrate that it is necessary to include easy examples to ensure coverage in high-density areas of the data distribution, which leads to the next factor of consideration i.e. data diversity.

**Data diversity.** Representation structure has been explored in several works as the key to the generalization of deep learning models; variance in representations for each class should be as large as possible while also being uncorrelated from other classes (Xia et al., 2023). The diversity of a dataset can be captured in many ways such as coding rate (Yu et al., 2020; Chan et al., 2022), max dispersion or convex hull volume (Yu et al., 2022) and coverage (Sener & Savarese, 2018; Zheng et al., 2022). A set $S'$ is a $r$-cover of another set $S$, when a set of $r$-radius balls centered at each element in

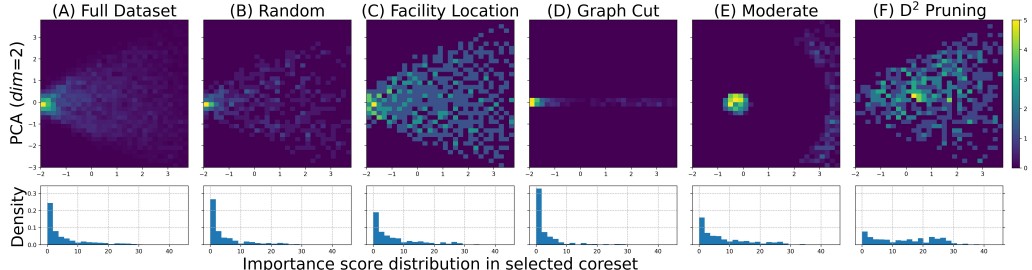

Figure 2: Sampling Methods. Density heat map of data distribution (left) and histogram of importance scores (right) in (A) a single class in the CIFAR10 dataset, and coresets selected under 90% pruning rate via (B) random sampling, diversity-only submodular approaches (C) facility location, (D) graph-cut (Iyer et al., 2021), (E) Moderate selection (Xia et al., 2023) and (F) our method, $\mathbb{D}^2$ PRUNING, designed to balance data diversity (pretrained ResNet18 features) and difficulty (Toneva et al., 2018).

$S'$ covers the entire $S$. The radius $r$ can be used as a metric to measure coverage of $S'$ on $S$ (Sener & Savarese, 2018). Zheng et al. (2022) introduce the metric $\text{AUC}_{pr}$ (Area under coverage), which is computed against test set $D_{test}$ i.e. $\text{AUC}_{pr}(S) = E_{x \in D_{test}}[min_{x' \in S} d(x', x)]$ and theoretically show that it is important to minimize the $\text{AUC}_{pr}$ for better generalization. Difficult samples tend to be rarer samples found in the low-density areas of the data distribution whereas easy samples tend to lie in high-density areas. An effective coreset should contain sufficient samples from both areas to ensure maximum coverage. However, optimizing for diversity only leads to coresets with a skewed distribution over example difficulty. As we show in Fig. 2(c), $k$-center selection minimizes the distance of samples in $S$ from $S'$ and has high coverage of the underlying data distribution. But, the selected coreset contains a disproportionate number of easy samples, rendering it ineffective.

*Example difficulty* and *diversity* are two complementary factors that make an effective coreset. Hence, coreset selection methods need to unify the influence of these factors in a constructive manner. To this end, we represent the dataset $S$ as a graph and introduce a novel message-passing algorithm (Vashishth et al., 2019a;b), $\mathbb{D}^2$ PRUNING, that accounts for both factors when selecting samples for coreset.

## 3 $\mathbb{D}^2$ PRUNING: MESSAGE PASSING FOR CORESET SELECTION

Consider a dataset $\mathcal{S}$, where each sample $s$ is represented in an embedding space, i.e., $s \in \mathbf{R}^d$. We seek to select a coreset $S'$ consisting of a subset of the samples in $\mathcal{S}$ as outlined in Sec. 2.1. Moreover, our goal is to combine the influence of embedding distance and difficulty scores when selecting samples for coreset (see Sec. 2.2). This setting naturally lends itself to a representation using undirected graph $\mathcal{G}$, where each sample is represented as a node with node-feature $x_i$, and edge weights $e_{ij}$ to indicate its connectivity with other samples in the embedding space (see Fig. 1(a)). We use message-passing to 'inform' a sample about (a) its proximity to adjacent samples in an embedding space, and (b) the difficulty scores of its neighbors. First, we briefly discuss message passing for graphs, and then we discuss our proposed algorithm, $\mathbb{D}^2$ PRUNING.

### 3.1 MESSAGE PASSING

Message passing (Hamilton et al., 2017) is a widely-used operation performed on graphs to propagate information from a node's neighbors to itself and update the state of the node based on the newly acquired information. For instance, Gilmer et al. (2017); Gasteiger et al. (2020) use message-passing to encode molecular structures for chemical prediction. The message-passing phase is defined in terms of a message function $M$ and a node update function $U$. In the message passing phase, a given node $i$ receives messages from each of its neighbors and aggregates them to update its feature value:

$$m_i = \sum_{j \in \mathcal{N}(i)} m_{ij} ; \quad \text{where} \ m_{ij} = M(x_j, e_{i,j}) \tag{2}$$

$$x_i = U(x_i, m_i) \tag{3}$$

where $\mathcal{N}(i)$ denotes the neighbors of node $i$ in graph $\mathcal{G}$. $U$ is an aggregation function that accounts for the messages received from all neighbors, as well as the node's own feature.

## 3.2 $\mathbb{D}^2$ PRUNING

$\mathbb{D}^2$ PRUNING consists of 3 stages i.e., (a) Graph initialization, (b) forward message passing, and (c) data selection via reverse message passing.

**Graph initialization.** We create a single, sparse graph for the dataset $S$ where each sample in $S$ is represented by a node $i$ in the graph. In order to account for example difficulty during coreset selection, we initialize the node feature as the difficulty score of the sample based on training dynamics of the model $\theta$ trained on $S$, i.e., $x_i = f_\theta(s_i)$, where $f(.)$ is the scoring function. In practice, the scoring function can be one of the many metrics used to measure difficulty such as forgetting (Toneva et al., 2018), consistency score (Jiang et al., 2021), and self-supervised metrics like prototypicality (Sorscher et al., 2022) etc. Next, we collect the $k$ nearest neighboring samples for every sample in the dataset. Within the graph, the connecting edges between each node $i$ and its $k$ nearest neighbors are initialized with a non-zero edge weight $e_{i,j}$, where node $j$ is one of the $k$ nearest neighbors (see Fig. 1(a)). All other edge weights are set to zero, leading to a sparse graphical representation of the entire dataset $S$. The edge weight $e_{i,j}$ represents the proximity of the two nodes $i,j$ using the RBF kernel of the distance $d(i,j)$. We use the Euclidean distance as the distance function i.e., $d(i,j) = ||v_i - vj||$ where $v_i$ is the embedding vector for sample $i$.

**Forward message passing.** In this step, each node $i$ in the graph receives information about its neighborhood via a single step of message propagation. Every connected node $j$ sends a message $M$ to node $i$ about its importance score which is scaled by the edge weight as,

$$M(x_j, e_{ij}) = e_{i,j} * x_j \; ; \quad \text{where } e_{i,j} = \exp\left(-\gamma_f * d(i,j)^2\right) \tag{4}$$

The intuition behind this definition is that samples that are farther away from the node but are of higher difficulty should be weighted similarly to samples that are closer to the node and have lower difficulty. This promotes diversity in the coreset by ensuring representation from all regions of the data distribution. Finally, the receiving node $i$ aggregates all of the messages received from its neighboring nodes and updates its own feature value as,

$$U_f(x_i, m_i) = x_i + \sum_{j \in \mathcal{N}(i)} M(x_j, e_{i,j}) \tag{5}$$

This reinforces the importance of dense regions comprising easy samples or sparse regions comprising difficult samples. Existing methods (Ash et al., 2019; Das et al., 2023) do not make a distinction between easy-to-learn and hard-to-learn areas in the data representation space whereas, this step in $\mathbb{D}^2$ PRUNING increases the importance of a sample by an amount that is proportional to the importance scores of the samples surrounding it, thus ranking an easy sample in a hard-to-learn area higher than that in an easy-to-learn area. Therefore, in this way, we start with a graph $\mathcal{G}$ where connectivity is based on the distance between two samples in the feature space and convert it into a graph based on distance as well as difficulty scores via message passing.

**Data selection via reverse message passing.** In the final step, samples in $S$ are ranked according to their corresponding *updated* node feature values in $\mathcal{G}$. Iteratively, the highest ranking sample $x_k = \arg\max_{i \in S} x_i$ is selected (Ebert et al., 2012), and its neighboring nodes are down-weighted to maximize the diversity of the coreset. However, since the distance between two nodes is a representation of their semantic similarity, neighboring nodes that are farther away from the selected node must be down-weighted relatively less than those that are closer. We implement this via reverse message passing, where the neighboring nodes receive a weighted message from the selected node and use it to update their feature value as,

$$x_j = x_j - e_{k,j} * x_k, \;\; \forall j \in \mathcal{N}(k) \; ; \quad \text{where } e_{k,j} = \exp\left(-\gamma_r * d(k,j)^2\right), \tag{6}$$

where a lower value of $\gamma_r$ causes larger updates in connected nodes and vice-versa. With these steps, $\mathbb{D}^2$ PRUNING selects a coreset that contains samples from all regions of the data distribution and are more uniformly distributed over the range of difficulty scores (see Fig. 2(f)).

## 4 EXPERIMENTAL SETUP

**Tasks, Models & Datasets.** We evaluate $\mathbb{D}^2$ PRUNING on three vision datasets i.e., **CIFAR10**, **CIFAR100** (Krizhevsky et al., 2009) and **Imagenet-1K** (Deng et al., 2009), and two NLP datasets

i.e., a subset (2k train examples) of **ImDB** reviews for sentiment analysis, and the Adversarial NLI (**ANLI**) dataset (Nie et al., 2020) for natural language inference. To the best of our knowledge, we are the first to perform a systematic evaluation of coreset selection methods on NLP datasets. We evaluate unsupervsied $\mathbb{D}^2$ PRUNING on the DataComp (small) dataset (Gadre et al., 2023). We use ResNet-18 for CIFAR10 and CIFAR100, ResNet-34 for ImageNet-1K and RoBERTa for NLP datasets.

**Baselines.** (*Supervised*) We compare $\mathbb{D}^2$ PRUNING with several score-based and geometry-based coreset selection methods derived from the training dynamics of a model trained on the full dataset: A) **Random** selection. B) **Entropy** (Coleman et al., 2019) of prediction vector. C) **Forgetting** (Toneva et al., 2018) score. D) **EL2N** (Paul et al., 2021) i.e. L2 norm of error vectors. E) **Area under the margin** (Pleiss et al., 2020) score. E) **Moderate** (Xia et al., 2023) coreset consisting of samples at median distance from class center, F) **CCS** (Zheng et al., 2022) divides a range of difficulty scores into equal-sized bins and randomly samples from each bin, G) **CCS + k-Center**, where k-center samples are selected within each CCS bin, H) **BADGE** (Ash et al., 2019) that selects samples using k-means++ in the gradient vector space, I) **GLISTER** (Killamsetty et al., 2021b) uses bi-level optimization to select robust coresets, J) **CAL-SDS2** (Das et al., 2023) combines a facility location submodular function (Iyer et al., 2021) with forgetting scores, and J) **INGENIOUS** (Renduchintala et al., 2023), a diversity-only approach using facility location function for NLP tasks. (*Unsupervised*) We use self-supervised embeddings to compare $\mathbb{D}^2$ PRUNING with A) **Prototypicality** (Sorscher et al., 2022) computes k-means clusters using embeddings and selects samples farthest from cluster center, B) **CCS over prototypicality scores**, and C) **Moderate** selection (Xia et al., 2023).

**Implementation.** In the supervised approach of $\mathbb{D}^2$ PRUNING, graph nodes are initialized with supervised difficulty score values and feature embeddings extracted from the model trained on the entire dataset. We use the forgetting score for CIFAR10, CIFAR100 and AUM score for ImageNet-1K (Zheng et al., 2022). We substitute the forgetting score with variance (Swayamdipta et al., 2020) for NLP datasets since they are trained for fewer epochs and the [CLS] token representation in RoBERTa models for feature embeddings. Self-supervised $\mathbb{D}^2$ PRUNING is initialized with feature embeddings from SwAV (Caron et al., 2020) for ImageNet-1K and uniform difficulty scores over the dataset.

**Computational Complexity of $\mathbb{D}^2$ PRUNING.** Graph initialization involves getting the $k$-nearest neighbors which are computed on a A100 GPU using PyTorch, taking $<2$ minutes for CIFAR10, CIFAR100, Adversarial NLI and ImDB datasets, and approx. 12 minutes for ImageNet-1K at quadratic time complexity $\mathcal{O}(vn^2)$, where $v$ is the vector dimension. We use faiss indexing (CPU) to get the *approximate* nearest neighbors for the 12.8 M samples in the Datacomp dataset taking nearly 55 minutes (8 workers) at $\mathcal{O}(d\log(d))$ time complexity, where $d$=256K is the number of documents in the faiss index (Johnson et al., 2019). Forward message passing is a parallelizable step of linear time complexity that scales with $k$ as $\mathcal{O}(nk)$. The iterative selection step in $\mathbb{D}^2$ PRUNING takes $\mathcal{O}(n)$ time in our optimized implementation, completing in $<5$ minutes for DataComp (Sec. B, Appendix).

**Algorithm Hyperparameters.** We use the best reported hyperparameters for baseline methods. For $\mathbb{D}^2$ PRUNING, we set the forward message passing weight $\gamma_f$ to 1.0 and perform a sweep over $k = \{1, 5, 10, 15\}$ and $\gamma_r = \{0, 0.1, 0.2...1.0\}$ for CIFAR10, CIFAR100 datasets. Insights from these runs are used to select three configurations for each run on ImageNet-1K (see Sec. 5.2).

# 5 RESULTS & DISCUSSION

## 5.1 COMPARISON TO SUPERVISED CORESET SELECTION METHODS

We evaluate $\mathbb{D}^2$ PRUNING and other coreset selection methods outlined in Sec. 4 on three vision datasets and present results in Tab. 1. $\mathbb{D}^2$ PRUNING demonstrates consistent gains over the previous state-of-art for all datasets at low and medium pruning rates. $\mathbb{D}^2$ PRUNING yields significant gains ($p < 0.05$) i.e., 1.0% and 1.4%, over the previous best for 50% and 80% pruning rates on ImageNet-1K, showing the efficacy of graphs and message passing for coreset selection.[3] Notably, random pruning works surprisingly well for ImageNet-1K, especially for low pruning rates, and is hard to beat. CCS (Zheng et al., 2022) remains a strong baseline for 90% pruning rate and only benefits a little from additional diversity-based selection within the CCS bins (see CCS + k-Center in Tab. 1). CCS enforces a uniform distribution of sample difficulty scores in the coreset, which is beneficial at

---

[3]Statistical significance computed by bootstrapping 100K samples (Noreen, 1989; Tibshirani & Efron, 1993)

Table 1: Results on Vision Datasets. Performance (acc.) of $\mathbb{D}^2$ PRUNING and baselines on CIFAR10, CIFAR100 using ResNet18, and ImageNet-1k using ResNet34 models. Higher is better.

| Dataset ($\rightarrow$) | CIFAR10 | | | | | | CIFAR100 | | | | | | ImageNet-1K | | | | | |
|---|---|---|---|---|---|---|---|---|---|---|---|---|---|---|---|---|---|---|
| Pruning Rate ($\rightarrow$) | 0% | 30% | 50% | 70% | 80% | 90% | 0% | 30% | 50% | 70% | 80% | 90% | 0% | 30% | 50% | 70% | 80% | 90% |
| Random | 95.5 | 94.3 | 93.4 | 90.9 | 88.0 | 79.0 | 78.7 | 74.6 | 71.1 | 65.3 | 57.4 | 44.8 | 73.1 | 72.2 | 70.3 | 66.7 | 62.5 | 52.3 |
| Entropy (Coleman et al., 2019) | - | 94.8 | 92.9 | 90.1 | 84.1 | 72.1 | - | 74.7 | 68.9 | 60.3 | 49.6 | 35.0 | - | 72.3 | 70.8 | 64.0 | 55.8 | 39.0 |
| Forgetting (Toneva et al., 2018) | - | 95.7 | 94.9 | 88.1 | 73.8 | 46.3 | - | 76.0 | 68.1 | 49.3 | 30.3 | 20.6 | - | 72.6 | 70.9 | 66.5 | 62.9 | 52.3 |
| EL2N (Paul et al., 2021) | - | 95.4 | 94.8 | 89.2 | 78.6 | 30.3 | - | 75.6 | 68.1 | 47.2 | 24.8 | 11.8 | - | 72.2 | 67.2 | 48.8 | 31.2 | 12.9 |
| AUM (Pleiss et al., 2020) | - | 95.6 | 95.1 | 87.9 | 68.0 | 40.0 | - | 75.0 | 67.9 | 40.1 | 26.4 | 13.1 | - | 72.5 | 66.6 | 40.4 | 21.1 | 9.9 |
| GLISTER (Killamsetty et al., 2021b) | - | 95.1 | 94.5 | 90.9 | 85.8 | 69.3 | - | 78.1 | 74.1 | 68.2 | 58.1 | 52.4 | - | 68.7 | 65.6 | 61.4 | 60.3 | 52.0 |
| CAL-SDS2 (Das et al., 2023) | - | 95.7 | 94.4 | 92.1 | 88.9 | 84.6 | - | 77.6 | 74.5 | 69.1 | 64.7 | 56.2 | - | 71.8 | 70.5 | 68.0 | 64.2 | 56.3 |
| Moderate (Xia et al., 2023) | - | 93.9 | 92.6 | 90.6 | 87.3 | 81.0 | - | 74.6 | 71.1 | 65.3 | 58.5 | 45.5 | - | 72.0 | 70.3 | 65.9 | 61.3 | 52.1 |
| CCS (Zheng et al., 2022) | - | 95.4 | 95.0 | 93.0 | 91.0 | 86.9 | - | 77.1 | 74.4 | 68.9 | 64.0 | 57.3 | - | 72.3 | 70.5 | 67.8 | 64.5 | 57.3 |
| CCS + k-Center | - | 95.4 | 95.1 | 92.9 | 91.1 | 86.8 | - | 77.2 | 74.6 | 69.3 | 64.5 | 57.1 | - | 72.5 | 70.6 | 68.0 | 64.5 | 57.2 |
| BADGE (Ash et al., 2019) | - | 94.0 | 92.1 | 90.7 | 88.1 | 82.5 | - | 74.7 | 71.8 | 65.2 | 58.9 | 47.8 | - | 71.7 | 70.4 | 65.8 | 61.7 | 53.4 |
| $\mathbb{D}^2$ PRUNING | - | 95.7 | 94.9 | 93.3 | 91.4 | 87.1 | - | 78.2 | 75.9 | 70.5 | 65.2 | 56.9 | - | 72.9 | 71.8 | 68.1 | 65.9 | 55.6 |

Table 2: Results on NLP Datasets. Comparison of performance (acc.) of $\mathbb{D}^2$ PRUNING with existing coreset selection methods on ANLI, ImDB reviews using pretrained RoBERTa$_{\text{Large}}$. Higher is better.

| Dataset ($\rightarrow$) | Adversarial NLI (ANLI) | | | | | | ImDB Reviews (2k) | | | | | |
|---|---|---|---|---|---|---|---|---|---|---|---|---|
| Pruning Rate ($\rightarrow$) | 0% | 30% | 50% | 70% | 80% | 90% | 0% | 30% | 50% | 70% | 80% | 90% |
| Random | 48.8 | 46.3 | 45.2 | 43.6 | 42.8 | 40.3 | 91.8 | 91.2 | 91.12 | 90.4 | 84.6 | 81.3 |
| Entropy (Coleman et al., 2019) | - | 48.9 | 45.8 | 43.6 | 42.4 | 34.0 | - | 90.6 | 90.4 | 52.8 | 60.1 | 51.3 |
| Variance (Swayamdipta et al., 2020) | - | 48.3 | 45.4 | 41.7 | 40.1 | 38.7 | - | 91.4 | 91.0 | 90.2 | 51.5 | 50.7 |
| EL2N (Paul et al., 2021) | - | 47.7 | 46.3 | 43.9 | 41.1 | 40.3 | - | 91.6 | 91.4 | 51.0 | 50.6 | 50.3 |
| AUM (Pleiss et al., 2020) | - | 47.9 | 46.2 | 42.7 | 41.0 | 39.6 | - | 91.6 | 91.6 | 53.4 | 50.3 | 50.3 |
| GLISTER (Killamsetty et al., 2021b) | - | 48.6 | 46.2 | 43.8 | 43.1 | 39.9 | - | 90.9 | 91.2 | 90.1 | 89.1 | 87.4 |
| CAL-SDS2 (Das et al., 2023) | - | 48.7 | 46.8 | 44.1 | 43.1 | 40.2 | - | 90.7 | 90.5 | 85.4 | 86.2 | 88.3 |
| Moderate (Xia et al., 2023) | - | 46.1 | 44.5 | 43.2 | 42.8 | 40.3 | - | 91.4 | 91.2 | 90.9 | 89.8 | 85.4 |
| CCS (Zheng et al., 2022) | - | 48.5 | 46.2 | 44.5 | 43.2 | 40.4 | - | 91.6 | 90.8 | 90.2 | 89.6 | 87.5 |
| CCS + k-Center | - | 48.4 | 46.3 | 44.1 | 43.2 | 40.2 | - | 91.4 | 91.0 | 90.6 | 90.2 | 88.2 |
| BADGE (Ash et al., 2019) | - | 47.3 | 45.8 | 44.0 | 43.1 | 39.5 | - | 91.3 | 90.9 | 90.0 | 90.1 | 89.5 |
| INGENIOUS (Renduchintala et al., 2023) | - | 44.3 | 46.1 | 43.8 | 41.1 | 40.3 | - | 91.1 | 87.6 | 89.5 | 87.8 | 82.4 |
| $\mathbb{D}^2$ PRUNING | - | 48.9 | 46.7 | 45.3 | 44.5 | 40.3 | - | 91.7 | 91.6 | 91.2 | 90.9 | 90.3 |

high pruning rates for providing even coverage over easy and difficult samples. However, at lower pruning rates (or with increasing data budget), difficult training samples yield a lower test loss from deep learning models (Sorscher et al., 2022). The hyperparameters $k$ and $\gamma_r$ in $\mathbb{D}^2$ PRUNING (see Sec. 3) allow flexibility in the distribution of easy/difficult samples in coresets. We find that higher values of $\gamma_r$ and lower value of $k$ in $\mathbb{D}^2$ PRUNING leads to a coreset that is skewed towards more difficult samples and benefits performance at lower pruning rates. Conversely, low $\gamma_r$ and high $k$ lead to an equitable distribution over easy/difficult samples and are more useful for higher pruning rates. CAL-SDS2 (Das et al., 2023) also introduces a tunable hyperparameter for balancing difficulty and diversity, however, its use of facility location (Iyer et al., 2021) for measuring diversity yields lower gains than the graph-based local neighborhoods in $\mathbb{D}^2$ PRUNING. See discussion on hyperparameters in Sec.5.2 and qualitative analysis of coresets in Sec. D, Appendix.

Results from the evaluation of various coreset selection methods, including $\mathbb{D}^2$ PRUNING, on NLP datasets are presented in Tab. 2. First, we find that when pretrained language models (PLMs) are finetuned on task-specific datasets, the models *do not* suffer from a catastrophic decline in performance at high pruning rates, in contrast to models trained from scratch on vision datasets. For IMDB reviews, the performance of finetuned RoBERTa goes from 91.8% at 0% pruning to 81.3% at 90% pruning using random sampling. The performance improves to 87.5% using CCS coreset selection and further improves to 90.3% ($p < 0.05$) using $\mathbb{D}^2$ PRUNING. The ANLI dataset has been carefully crafted with an iterative, adversarial human-and-model-in-the-loop process, and hence, is significantly less redundant than conventional NLP datasets. The performance for ANLI falls from 48.8% to 42.8% at 80% pruning using random sampling. In this case, CCS coreset selection does not lead to a significant improvement in performance (43.2%), whereas $\mathbb{D}^2$ PRUNING improves the performance by 1.7% to obtain 44.5% ($p < 0.05$). Score-based selection methods largely fail to yield results better than random pruning at high pruning rates. Additionally, the use of facility location function for representing diversity in CAL-SDS2 (Das et al., 2023) and INGENIOUS (Renduchintala et al., 2023) yield less gains than our graph-based approach in $\mathbb{D}^2$ PRUNING.

## 5.2 ANALYSIS OF $\mathbb{D}^2$ PRUNING

$\mathbb{D}^2$ PRUNING contains two hyperparameters, $k$ nearest neighbors and reverse message passing weight $\gamma_r$ (see Sec. 3) that allow various distributions of importance scores in the selected coreset. At low

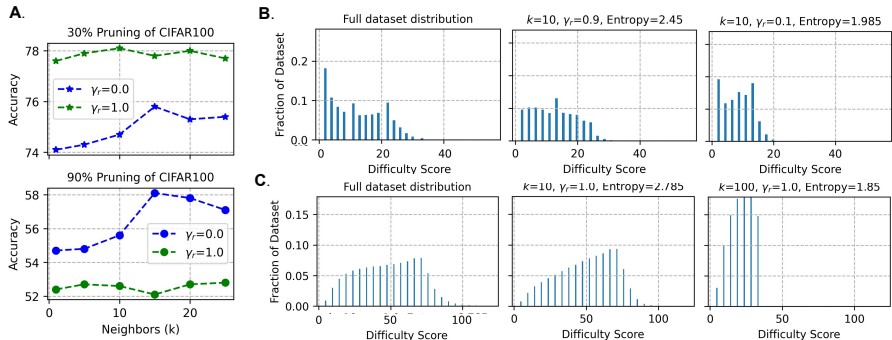

Figure 3: Effect of $k$, $\gamma_r$. (A) Accuracy at 30%, 90% pruning of CIFAR100 for nearest neighbors ($k$) and message passing weight $\gamma_r$ values; Distribution of difficulty scores in the best coresets selected via $\mathbb{D}^2$ PRUNING for 30% (center) and 70% (right) pruning of (B) CIFAR100, (C) ImageNet-1K.

pruning rates (see top, Fig. 3(a)), higher $k$ has a small effect on performance when the updates during reverse message passing are weak ($\gamma_r$=1.0). However, the coresets selected at high $k$ and low $\gamma_r$ include a majority of the difficult samples from the full dataset, which works best for low pruning rates on CIFAR100, as demonstrated by the distribution of importance scores in best-performing coreset at 30% pruning rate (see Fig. 3(B), center). We use this insight to pick a similar configuration of $\mathbb{D}^2$ PRUNING for ImageNet-1K and find that it transfers well. The distribution of difficulty scores in the best-performing coreset of ImageNet-1K at 30% pruning rate is presented in Fig. 3(C).

Higher $k$ improves performance when large updates ($\gamma_r$=0.0) are being made to the nodes connected to the selected node at high pruning rates (see bottom, Fig. 3(a)). This is because low $\gamma_r$ value leads to aggressive downweighting of semantically similar samples when a sample is selected and promotes diversity under a fixed data budget. The selected samples also form an equitable distribution over a small range of difficulty scores. Consequently, such coresets work best for medium-to-high pruning rates, as evidenced by the distribution of difficulty scores in the best performing coresets at 70% pruning rate for CIFAR100 and ImageNet-1K (see Fig. 3(B,C), right).

### 5.3 SELF-SUPERVISED AND UNSUPERVISED APPROACHES USING $\mathbb{D}^2$ PRUNING

Existing methods for obtaining sample difficulty scores generally rely on a model trained on the full dataset, which undermines their utility for scalably curating new datasets. Hence, we adopt $\mathbb{D}^2$ PRUNING for self-supervised and unsupervised data selection approaches.

**Self-supervised coreset selection.** Sorscher et al. (2022) use embeddings from SwAV, a model trained on ImageNet-1k in a self-supervised manner, and use the spatial distribution of the samples in the embedding space to assign difficulty scores (prototypicality). This menthod suffers drastically at over 30% pruning rates (see Fig. 4.). When combined with CCS, it yields 10% gain for 90% pruning rate and lesser gains for 70%, 80% pruning rates. We adopt $\mathbb{D}^2$ PRUNING for a similar self-supervised approach by using SwAV embeddings to compute sample distances and initialize node features with a unit value. In the absence of difficulty scores, $\mathbb{D}^2$ PRUNING ranks the samples solely by the density of their neighborhood in the embedding space. $\mathbb{D}^2$ PRUNING improves performance by 3% at 80% pruning rate and provides similar gains over prototypicality for lower pruning rates.

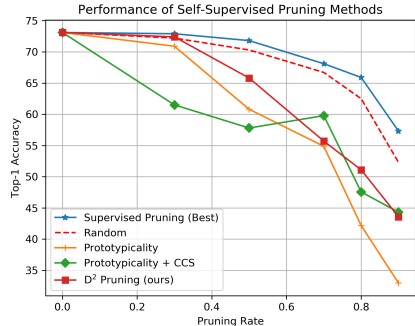

Figure 4: Self-supervised pruning of ImageNet-1K. $\mathbb{D}^2$ PRUNING performs as well as supervised pruning at 30% and significantly improves over existing methods.

**Unsupervised data filtering.** Gadre et al. (2023) show that a simple strategy of retaining the samples with a high CLIP score is a strong baseline filtering method (see Tab. 3) on DataComp, a massive unfiltered corpus of images and texts to train CLIP-style models (Radford et al., 2021).[4] However, a

---

[4]Our reproduced numbers are lower than Gadre et al. (2023) because some images in the original corpus fail download. We report improvements using $\mathbb{D}^2$ PRUNING on this subset of images for fair comparison.

Table 3: Results on DataComp. Comparison of performance (acc.) of $\mathbb{D}^2$ PRUNING with CCS (Zheng et al., 2022) and data filtering methods presented in Gadre et al. (2023). Higher is better.

| Filtering Strategy | Dataset Size | ImageNet | ImageNet Dist. Shift | VTAB | Retrieval | Average |
|---|---|---|---|---|---|---|
| No filtering (Gadre et al., 2023) | 12.8M | 2.5 | 3.3 | 14.5 | 11.4 | 13.2 |
| Text-based filtering (Gadre et al., 2023) | 3.2M | 4.6 | 5.2 | 16.9 | 12.5 | 15.7 |
| Image-based filtering (Gadre et al., 2023) | 3.2M | 4.3 | 4.7 | 17.8 | 12.1 | 15.9 |
| CLIP score (L/14 30%) (Gadre et al., 2023) | 3.8M | 5.1 | 5.5 | 19.0 | 11.7 | 17.3 |
| CLIP score (L/14 30%, reproduced) | 3.8M | 5.1 | 5.6 | 17.0 | 11.9 | 16.0 |
| CCS (Zheng et al., 2022) | 3.8M | 2.6 | 3.7 | 14.3 | **14.2** | 13.8 |
| $\mathbb{D}^2$ PRUNING (image + text) | 3.8M | 5.1 | **5.6** | 18.2 | 11.7 | 17.0 |
| $\mathbb{D}^2$ PRUNING (image only) | 3.8M | 4.4 | 5.1 | 16.9 | 12.1 | 15.9 |
| $\mathbb{D}^2$ PRUNING (text only) | 3.8M | 4.9 | 5.5 | 17.0 | 12.3 | 16.6 |
| T-MARS (Maini et al., 2023) | 2.5M | 6.3 | 6.6 | 17.9 | 12.8 | 17.7 |
| T-MARS + $\mathbb{D}^2$ PRUNING (image + text) | 2.5M | **6.5** | **6.7** | **19.1** | 12.8 | **18.8** |

strategy based on individual sample scores only ignores potential redundancies in the dataset and may allot unnecessary data budget to an easy but dense region of the sample space. Hence, we adapt $\mathbb{D}^2$ PRUNING for filtering DataComp by treating the CLIP score as the difficulty score and using CLIP embeddings for computing sample distances. The data selected by $\mathbb{D}^2$ PRUNING using both, CLIP text and image embeddings, for computing sample distances improves average zero-shot performance on 38 image classification, multimodal datasets by 1% at same data budget (see Tab. 3.). When combined with filtering method T-MARS (Maini et al., 2023) that removes images containing overlapping textual content (with caption), $\mathbb{D}^2$ PRUNING achieves cumulative improvements.

## 6 RELATED WORK

**Coreset Selection.** Coreset selection has been widely studied in machine learning (Welling, 2009; Chen et al., 2010; Feldman et al., 2011) for supervised learning. Recent works have focused on large datasets and deep networks. Geometry-based methods remove redundant information (Welling, 2009; Sener & Savarese, 2018). Uncertainty/loss/error-based methods estimate the difficulty of a sample from model confidence (Swayamdipta et al., 2020) or its training dynamics (Toneva et al., 2018; Paul et al., 2021; Bachem et al., 2015). Submodular functions (Wei et al., 2015; Killamsetty et al., 2023), gradient-matching (Mirzasoleiman et al., 2020), and optimization (Yang et al., 2022; 2023; Park et al., 2022) have been explored for coreset selection. Zhou & Bilmes (2018); Zhou et al. (2020); Das et al. (2023) combine submodular functions with difficulty scores for selecting data. Joshi & Mirzasoleiman (2023) study the importance of data samples for self-supervised learning via submodular optimization. We combine data diversity and sample difficulty via graphs for selection.

**Data Pruning in NLP**. Works exploring coreset selection methods for NLP datasets have been far and few (Fayyaz et al., 2022). Abbas et al. (2023) removes semantic duplicates from C4 dataset (Raffel et al., 2020) to reduce data size and improve performance. Kaddour (2023) introduce a small version of the Pile dataset (Gao et al., 2020) for pretraining BERT (Devlin et al., 2018; Liu et al., 2019). We evaluate coreset selection methods on sentiment analysis, natural language inference tasks.

**Graphs & Message Passing for Data Selection.** Neural message passing (Yadav et al., 2019; Yadati et al., 2019) is well-explored in graph neural networks for chemical structures (Gilmer et al., 2017), however, has seen less exploration in the representation of datasets. Kim et al. (2021) use message-passing to learn the topology of data in online learning. Ebert et al. (2012) use message-passing based on feature distance only for performing graph-based density sampling during active learning. Hongjin et al. (2022) construct a sparse graph from the $k$-nearest neighbors of in-context examples and down-weight the selected example's connected nodes, which is similar to the reverse message-passing step in $\mathbb{D}^2$ PRUNING. In contrast, we initialize the nodes with sample importance scores and first use forward message-passing to merge the influence of importance score and density of local neighborhood to rank samples in $\mathbb{D}^2$ PRUNING.

## 7 CONCLUSION

We introduce a novel coreset selection algorithm, $\mathbb{D}^2$ PRUNING, based on message-passing within a graph representing the dataset. Our algorithm combines data diversity and difficulty to select a coreset that outperforms existing coreset selection methods at low-to-medium pruning rates on multiple vision and NLP benchmarks, and can be adapted into self-supervised, unsupervised data selection.

**Reproducibility.**    We report the training hyperparameters used for our best models, as well as the best hyperparameters for $\mathbb{D}^2$ PRUNING (see Sec. 3 and a discussion in Sec. 5.2) in the Appendix. The code for running the experiments in our paper is available as part of the supplementary submission. All datasets used in our experiments are openly available.

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

## OVERVIEW

The appendix is organized as follows:
**Section A**: Details of the datasets, baselines and the best hyperparameters for our models.
**Section B**: Computational complexity of $\mathbb{D}^2$ PRUNING for all datasets.
**Section C**: Additional results for high pruning rates and ablation experiments.
**Section D**: Qualitative analysis of coresets selected via $\mathbb{D}^2$ PRUNING.
**Section E**: Limitations and license.

## A  DATASETS & HYPERPARAMETERS

### A.1  DATASETS

**Vision Benchmarks.**  We use the CIFAR10, CIFAR100 (Krizhevsky et al., 2009) and ImageNet-1K (Deng et al., 2009) image classification datasets for our experiments on vision benchmarks. The CIFAR10 dataset consists of 60000 32x32 color images for 10 classes, with 6000 images per class. The training and test splits contain 50000 and 10000 images respectively. The CIFAR100 dataset has 100 classes containing 500 and 100 images per class in the training and test splits respectively. Details about the class labels in CIFAR10, CIFAR100 datasets can be found here. The ImageNet-1K dataset comprises approximately 1.2 million real-world images distributed over 1000 object classes. It contains 1,281,167 and 50,000 images in training and validation splits respectively.

**NLP Benchmarks.**  We select two popularly used NLP tasks i.e. natural language inference (NLI) (Bowman et al., 2015) and sentiment analysis (Turney, 2002). For natural language inference, we use the Adversarial NLI dataset (Nie et al., 2020) that has been created in an iterative human-and-model-in-the-loop adversarial procedure. During each iteration, human annotators are instructed to devise examples that the current best models are unable to answer correctly. The models are trained on these challenging annotations for stronger performance. Multiple rounds of such iterations result in a challenging NLI benchmark. We use the data created in the third (and final) round of this process which contains 100459, 1200, and 1200 examples in the training, development, and test splits respectively. We use the ImDB reviews dataset (Maas et al., 2011) for the sentiment analysis task. The original dataset contains 25000 examples each in the training and test splits and is a binary

Table 4: Best values of nearest-neighbors ($k$) and reverse message passing weight ($\gamma_r$) for vision datasets. See a discussion on these hyperparameters in Sec. 5.2.

| Dataset ($\rightarrow$) | CIFAR10 | | | | | | CIFAR100 | | | | | | ImageNet-1K | | | | | |
|---|---|---|---|---|---|---|---|---|---|---|---|---|---|---|---|---|---|---|
| Pruning Rate ($\rightarrow$) | 0% | 30% | 50% | 70% | 80% | 90% | 0% | 30% | 50% | 70% | 80% | 90% | 0% | 30% | 50% | 70% | 80% | 90% |
| Nearest Neighbors ($k$) | - | 10 | 5 | 1 | 2 | 2 | - | 10 | 10 | 10 | 5 | 15 | - | 50 | 50 | 100 | 10 | 10 |
| Reverse Message Passing ($\gamma_r$) | - | 0.9 | 1.0 | 0.1 | 0.0 | 0.0 | - | 0.9 | 0.8 | 0.3 | 0.3 | 0.0 | - | 1.0 | 1.0 | 0.3 | 0.1 | 0.0 |

Table 5: Best values of nearest-neighbors ($k$) and reverse message passing weight ($\gamma_r$) for NLP datasets and self-supervised $\mathbb{D}^2$ PRUNING of ImageNet-1K. See details in Sec. 5.2.

| Dataset ($\rightarrow$) | Adversarial NLI | | | | | | ImDB(2K) | | | | | | ImageNet-1K (self-supervised) | | | | | |
|---|---|---|---|---|---|---|---|---|---|---|---|---|---|---|---|---|---|---|
| Pruning Rate ($\rightarrow$) | 0% | 30% | 50% | 70% | 80% | 90% | 0% | 30% | 50% | 70% | 80% | 90% | 0% | 30% | 50% | 70% | 80% | 90% |
| Nearest Neighbors ($k$) | - | 15 | 10 | 5 | 5 | 5 | - | 10 | 10 | 10 | 5 | 2 | - | 50 | 100 | 25 | 10 | 25 |
| Reverse Message Passing ($\gamma_r$) | - | 1.0 | 1.0 | 0.1 | 0.1 | 0.0 | - | 1.0 | 0.8 | 0.3 | 0.0 | 0.0 | - | 1.0 | 1.0 | 0.5 | 0.5 | 0.0 |

classification dataset. Our experiments showed that models trained on 10% of this dataset achieved nearly the same performance as 100% of the dataset. We observed similar trends for other popular sentiment analysis benchmarks as well such as Yelp Reviews (Zhang et al., 2015), SST2 (Socher et al., 2013) etc. Hence, we created an in-house version of the ImDB Reviews dataset that contains 2000, and 1000 samples in the training and development splits respectively, that are randomly selected from the original training set. We retain the original test split containing 25000 samples for evaluation in our experiments.

## A.2 BASELINES

(*Supervised*) We compare $\mathbb{D}^2$ PRUNING with several score-based and geometry-based coreset selection methods derived from the training dynamics of a model trained on the full dataset as discussed in Zheng et al. (2022): A) **Random** selection of examples. B) **Entropy** (Coleman et al., 2019) of a model's prediction vector. C) **Forgetting** (Toneva et al., 2018) score for each example i.e., the number of times a model predicts the example incorrectly after having predicted correctly in the previous epoch. D) **EL2N** (Paul et al., 2021) i.e. L2 norm of error vectors. E) **Area under the margin** (Pleiss et al., 2020) score that measures the gap between the prediction probability of the correct target and the next highest probability target. E) **Moderate** coresets (Xia et al., 2023) that selects samples at median distance from class center, F) Coverage-based Coreset Selection (**CCS**) (Zheng et al., 2022) that divides a range of difficulty scores into equal-sized bins and randomly samples from each bin, and is state-of-art for high pruning rates, G) **CCS + k-Center**, where k-center samples are selected within each CCS bin, H) **BADGE** that selects diverse samples using k-means++ in the gradient vector space, I) **GLISTER** (Killamsetty et al., 2021b) uses bi-level optimization to select robust coresets, J) **CAL-SDS2** (Das et al., 2023) combines a facilty location submodular function (Iyer et al., 2021) with entropy scores to unify the effects of difficulty score and diversity, and J) **INGENIOUS** (Renduchintala et al., 2023), a diversity-only approach that uses facility location as the information gain function for NLP tasks. (*Unsupervised*) We compare $\mathbb{D}^2$ PRUNING with A) **Prototypicality** (Sorscher et al., 2022) that uses self-supervised embeddings to compute k-means clusters and treats samples at a farther distance from the cluster center as more important, B) **CCS over prototypicality scores**, and C) **Moderate** coreset selection (Xia et al., 2023) over the self-supervised embeddings.

## A.3 TRAINING HYPERPARAMETERS

**Coreset Selection.** We use the recommended hyperparameters in Zheng et al. (2022) for experiments using Coverage-based coreset selection (CCS) i.e. 50 bins (or *strata*) for all pruning rates. Models trained on vision datasets are also subjected to a hard cutoff rate $\beta$ on the difficulty score for eliminating outliers or erroneous samples (see Zheng et al. (2022) for the values). We report the best hyperparameters for $\mathbb{D}^2$ PRUNING in Tabs. 4 & 5.

**Models.** We follow the best training hyperparameters for ResNet18 model and ResNet34 models as suggested in Zheng et al. (2022) to remain cmoparable to the numbers reported in their work. For fine-tuning of pretrained RoBERTa on NLP datasets, we perform a grid search over learning

rates $\{1e^{-5}, 2e^{-5}, 5e^{-5}, 1e^{-4}\}$ and batch sizes $\{8, 16, 32\}$ using 100% of the data, which results in learning rate of $1e^{-4}$ and batch size of 32 for Adversarial NLI, ImDB (2k) datasets. Models are trained on pruned datasets using the same hyperparameters that are used for training 100% of the data. The maximum number of training steps is kept constant across all pruning rates. RoBERTa models are trained for 10000 and 1500 training steps for Adversarial NLI and ImDB (2k) datasets respectively, with early stopping.

---

**Algorithm 1** $\mathbb{D}^2$ PRUNING for Data Selection

---

 1: **if** `selection = supervised` **then**
 2:     **Input Data**: $\mathbf{D} = <x, y>$
 3: **else**
 4:     **Input Data**: $\mathbf{D} = <x>$
 5: **end if**
 6: **Validation Data**: $\mathbf{D}_{val} = <x, y>$
 7: **Input Data**: $\mathbf{D}_{test} = <x, y>$
 8:
 9: **Train:**
10: $\theta_t \leftarrow$ initialize trainable parameters
11: **for** $epoch = 1, 2, \ldots, N$ **do**
12:     Train $\theta_t$ on D
13: **end for**
14:
15: **Optimize Data Selection using $\mathbb{D}^2$ PRUNING:**
16: $\{\mathbf{k}\} \leftarrow$ grid search values for nearest-neighbor hyperparameter in $\mathbb{D}^2$ PRUNING
17: $\{\gamma_r\} \leftarrow$ grid search values for reverse message passing hyperparameter in $\mathbb{D}^2$ PRUNING
18: $|\mathbf{D}_s| \leftarrow$ number of samples to be selected
19: **for** $k$ in $\{\mathbf{k}\}$ **do**
20:     **for** $\gamma_r$ in $\{\gamma_r\}$ **do**
21:
22:         **Select $\mathbf{D}_s \subset \mathbf{D}$ using $\mathbb{D}^2$ PRUNING:**
23:         $\mathcal{G} \leftarrow$ initialize graph in $\mathbb{D}^2$ PRUNING using $k, \gamma_r, \theta_t$
24:         **for** $d = 1, 2, \ldots, D$ **do**
25:             Perform forward message passing
26:         **end for**
27:         **for** $i = 1, 2, \ldots, |\mathbf{D}_s|$ **do**
28:             Select sample with highest node feature and add to $\mathbf{D}_s$
29:             Downweight neighbors of selected sample
30:         **end for**
31:         Obtain labels for $\mathbf{D}_s$
32:
33:         **Train on $\mathbf{D}_s$:**
34:         $\theta_v \leftarrow$ initialize trainable parameters
35:         **for** $epoch = 1, 2, \ldots, N$ **do**
36:             Train $\theta_v$ on $\mathbf{D}_s$
37:         **end for**
38:         **Evaluate** $\theta_v$ on $\mathbf{D}_{val}$
39:
40:     **end for**
41: **end for**
42:
43: **Evaluate** best $\theta_v$ on $\mathbf{D}_{test}$

---

# B   COMPUTATIONAL COMPLEXITY OF $\mathbb{D}^2$ PRUNING

We divide the runtime of $\mathbb{D}^2$ PRUNING into: (1) **Graph creation** which includes graph initialization and forward message passing, (2) **Iterative selection** (see Sec. 3) and present results in Tab. 6 for 100% data selection of the various datasets used in our experiments. Numbers are rounded to the nearest minute. Runtime for iterative selection is proportional to the size of the coreset being selected. Hence, in practice, the runtime for iterative selection is even lower since we only select a subset of the data in our experiments. Time estimates of graph creation for different datasets are not strictly comparable because we run jobs of different batch sizes according to the size of the dataset to prevent

Table 6: Computational Overhead for $\mathbb{D}^2$ PRUNING. Comparison of runtime of $\mathbb{D}^2$ PRUNING for 100% selection of the various datasets in our experiments. $\mathbb{D}^2$ PRUNING can be divided into the 'Graph creation' and 'Iterative selection' steps (see General Response). Larger datasets like DataComp have a 'faiss indexing' step to enable fast nearest-neighbor lookup. Results are computed using a multi-thread implementation of $\mathbb{D}^2$ PRUNING using 8 workers on a CPU with 32 cores.

| Dataset ($\rightarrow$) | CIFAR10 | CIFAR100 | Adv. NLI | ImDB | DataComp | ImageNet-1K |
|---|---|---|---|---|---|---|
| **faiss indexing** | - | - | - | - | 25m | - |
| **Graph creation** | 2m | 1m | 4m | 1m | 30m | 15m |
| **Iterative selection** | 1m | 1m | 2m | 1m | 7m | 8m |
| **Total selection time** | 3m | 2m | 6m | 2m | 1h 2m | 23m |
| **Training time** | 4h 30m | 4h 45m | 2h | 15m | 4h 15m | 125h |

Table 7: Results on Vision Datasets. Comparison of performance (acc.) of $\mathbb{D}^2$ PRUNING with existing coreset selection methods for very high pruning rates on CIFAR10, CIFAR100 using ResNet18, and ImageNet-1k using ResNet34 models. Higher is better.

| Dataset ($\rightarrow$) | CIFAR10 | | | | | | CIFAR100 | | | | | | ImageNet-1K | | | | | |
|---|---|---|---|---|---|---|---|---|---|---|---|---|---|---|---|---|---|---|
| Pruning Rate ($\rightarrow$) | 0% | 90% | 95% | 99% | 99.5% | 99.9% | 0% | 90% | 95% | 99% | 99.5% | 99.9% | 0% | 90% | 95% | 99% | 99.5% | 99.9% |
| Random | 95.5 | 79.0 | 70.0 | 39.8 | 35.8 | 23.8 | 78.7 | 44.8 | 28.7 | 10.8 | 6.13 | 3.5 | 73.1 | 52.3 | 41.1 | 9.6 | 4.1 | 0.9 |
| CCS (Zheng et al., 2022) | - | 86.9 | **77.2** | 41.8 | 33.0 | **25.7** | - | **57.3** | **36.9** | 13.5 | **8.8** | **3.6** | - | **57.3** | **45.9** | **10.1** | 6.2 | 1.1 |
| $\mathbb{D}^2$ PRUNING | - | **87.1** | 74.5 | **44.4** | **33.8** | 24.6 | - | 56.9 | 35.8 | **14.2** | 5.9 | 2.4 | - | 55.6 | 44.8 | 7.6 | **7.2** | **1.9** |

OOM issues when computing similarity matrix. Additionally, we provide the approximate training times for each dataset computed on a single A100 GPU.

# C ADDITIONAL RESULTS

## C.1 RESULTS ON HIGH PRUNING RATES

We perform coreset selection at very high pruning rates (Guo et al., 2022) for CIFAR10, CIFAR100 and ImageNet-1K using $\mathbb{D}^2$ PRUNING and a select few baselines, and present results in Tab. 7. $\mathbb{D}^2$ PRUNING outperforms random selection as well as CCS (Zheng et al., 2022) in some scenarios, such as by 3% and 0.5% at 99% and 99.5% pruning of CIFAR10 respectively. However, we do not see any consistent trends in improvement using $\mathbb{D}^2$ PRUNING at very high pruning rates, especially for ImageNet-1K. Improvement margins using CCS also go down at high pruning rates, suggesting that diversity is not as important as difficulty when the data budget is extremely low (Sorscher et al., 2022).

## C.2 ABLATION EXPERIMENTS

**Multiple message passing iterations.** To better understand the effect of the forward message passing procedure in $\mathbb{D}^2$ PRUNING, we visualize a random subset of CIFAR10 samples in a 2-dimensional t-SNE (Van der Maaten & Hinton, 2008) embedding space in Fig. 5 before and after forward message passing under various scenarios. When the graph is first initialized in $\mathbb{D}^2$ PRUNING, the node feature is initialized with the sample's importance score (see Fig. 5A). A single iteration of a forward message passing over the local neighborhood of a sample consisting of $k$ nearest neighbors leads to the significant up-weighting of neighbors of a very important node. Thus, a higher $k$ leads up-weighting of a larger neighborhood of samples in the spatial dimension (see Fig. 5B vs. Fig. 5C). Consequently, the distribution of normalized node feature values has a heavier tail with increasing $k$, as compared to the distribution of original importance scores. In contrast, multiple iterations of message passing at the same $k$ have an effect similar to that of Gaussian smoothing in the embedding space (see Fig. 5D). With increasing iterations, the local neighborhood of a node becomes increasingly similar to that of other nodes in the graph, and hence, all nodes receive similar updates (see Fig. 5D vs. Fig. 5E). As a result, the distribution of node features is biased towards a narrow spectrum of values that no longer benefits the data selection task (see results in Tab. 8 in Appendix).

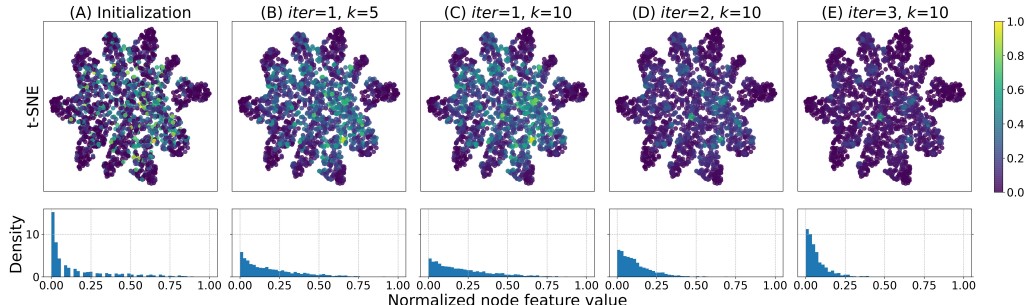

Figure 5: Effect of *forward* message passing iterations. (top) Scatter plots of CIFAR10 samples' *normalized* node feature values in a t-SNE embedding space ($dim$=2) and (bottom) corresponding histograms for the following scenarios: (A) Initialization of graph in $\mathbb{D}^2$ PRUNING, one-shot forward message passing with $k$=5 (B) and $k$=10 (C) nearest neighbors, and two-shot (D), three-shot (E) forward message passing at $k$=10.

Table 8: Ablation Analysis. Results from ablations of $\mathbb{D}^2$ PRUNING by varying the embedding representation (rows A-B), difficulty score (rows C-F), and multiple iterations of message passing (rows H-J) on CIFAR10 and Adversarial NLI datasets. Higher is better.

| Dataset (→) | | CIFAR10 | | | | | | Adversarial NLI | | | | | |
|---|---|---|---|---|---|---|---|---|---|---|---|---|---|
| Pruning Rate (→) | | 0% | 30% | 50% | 70% | 80% | 90% | 0% | 30% | 50% | 70% | 80% | 90% |
| A. | **Last Layer / [CLS]** | 95.5 | **95.7** | **94.9** | 93.3 | **91.4** | 87.1 | 48.8 | **48.9** | **46.7** | **45.3** | **44.5** | **40.3** |
| B. | **Pre-final convolutional / non-[CLS]** | - | 95.1 | **94.9** | 92.7 | 90.5 | 85.4 | - | 45.3 | 45.3 | 42.7 | 42.8 | 39.1 |
| C. | **Forgetting / Variance** | 95.5 | **95.7** | **94.9** | 93.3 | 91.4 | 87.1 | 48.8 | 48.9 | 46.7 | **45.3** | **44.5** | 40.3 |
| D. | **Entropy** | - | 95.6 | 94.6 | **93.8** | **91.9** | **87.3** | - | 48.5 | 46.1 | 45.4 | 44.2 | 40.1 |
| E. | **EL2N** | - | 94.9 | 94.2 | 93.1 | 91.1 | 86.0 | - | **49.3** | **47.9** | 45.2 | 44.3 | **40.3** |
| F. | **1-shot** | 95.5 | **95.7** | **94.9** | 93.3 | **91.4** | 87.1 | 48.8 | **48.9** | **46.7** | **45.3** | **44.5** | 40.3 |
| G. | **2-shot** | - | 94.0 | 94.1 | 89.9 | 89.2 | 85.6 | - | 47.4 | 45.2 | 44.9 | 43.1 | **40.4** |

**Effect of importance scores.** We experiment with different importance scores in $\mathbb{D}^2$ PRUNING and present results in Tab. 8 in Appendix. We find that the entropy score (Coleman et al., 2019) benefits performance on CIFAR10 at higher pruning rates, whereas EL2N (Paul et al., 2021) benefits performance on Adversarial NLI for low pruning rates. Importantly, we do not see large drops in performance with any of these score functions, suggesting that the idea of combining diversity and difficulty in $\mathbb{D}^2$ PRUNING is universally beneficial. Further, improved difficulty metrics can be paired with $\mathbb{D}^2$ PRUNING for larger improvements in data selection.

**Effect of embedding sources.** Next, we experiment with alternative sources of feature embeddings for measuring the distance between two samples. Since final layers in a task-specific model are known to be attuned to the task (Han & Tsvetkov, 2021), instead, we extract features from the last convolutional layer in ResNet18 for CIFAR10 and use the average of non-[CLS] tokens in RoBERTa for ImDB dataset (row B). We find that neither source is as effective as the features extracted from the last layer of the model trained on the full dataset. Especially, we see large drops in performance on the use of non-[CLS] token features for representing diversity. We leave the study of the utility of different embedding spaces for measuring diversity to future work.

## D ANALYSIS & DISCUSSION

**Qualitative analysis of coresets selected by $\mathbb{D}^2$ PRUNING**. In order to perform a qualitative analysis of the merits of $\mathbb{D}^2$ PRUNING, we first use the connectivity graph $\mathcal{G}$ to extract meaningful sub-populations from the entire ImageNet-1K dataset. For each sample, we recursively seek nearest neighbors that are situated at a distance in the embedding space that is less than a predefined threshold. Next, for each of these sub-populations, we differentiate the samples that appear in the coreset selected by $\mathbb{D}^2$ PRUNING at 30% pruning of ImageNet-1K. We present and analyze a few representative sub-populations in Fig. 6. First, we observe several cases where $\mathbb{D}^2$ PRUNING successfully avoids selecting perceptual duplicates (Abbas et al., 2023) in the coreset (see top left and middle left in

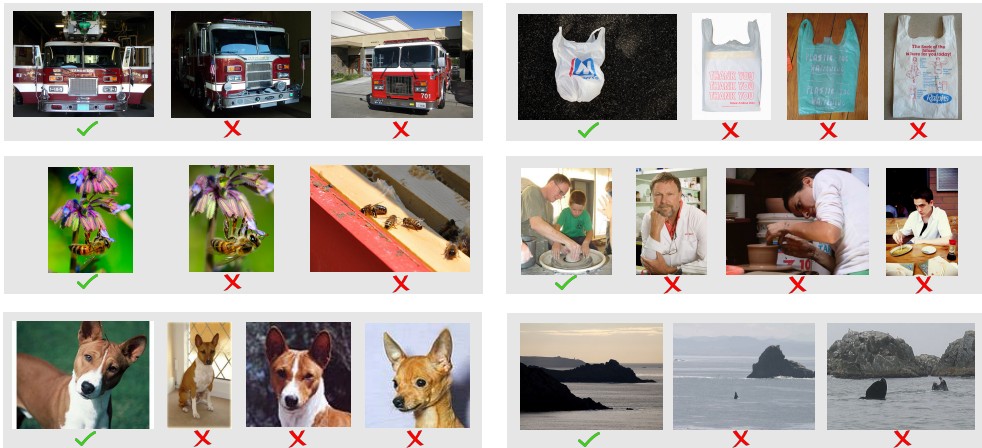

Figure 6: Example of coresets selected by $\mathbb{D}^2$ PRUNING from ImageNet-1K at 30% pruning rate. Image sub-populations are extracted from ImageNet-1K by a recursive traversal of the connectivity graph $\mathcal{G}$ initialized for $\mathbb{D}^2$ PRUNING. For each sub-population, we show the images retained in the coreset with ✓ and the images left out of the coreset with ✗.

Fig. 6). Next, we see multiple cases where a composite image is selected for the coreset, and images that contain one or more of the subjects/objects in the selected image are left out (see middle right in Fig. 6). Finally, we find that relying on the semantic similarity of pretrained embeddings can lead to the propagation of errors, as seen in the sub-population on the bottom right in Fig. 6. The images that contain dolphins are left out of the coreset because of their similarity to an image depicting a water landscape.

**Visualization of data distribution in coresets.** We showcase the results of various sampling methods for a single class in the CIFAR10 dataset in Fig. 2. The features are obtained from a ResNet18 network trained on the full training dataset and compressed to two dimensions using PCA ( 90% explained variance) for simpler visualization. As seen in Fig. 2(b), random sampling leads to relatively larger samples from the denser region of the distribution and consequently, a higher percentage of easy samples feature in the coreset after 90% pruning. By optimizing for diversity only via facility location (Iyer et al., 2021) submodular optimization (Fig. 2(c)), the diversity of the coreset remains high but it is plagued with the same problem as random sampling i.e. easier samples are preferred. Alternatively, the use of graph-cut function with cosine similarity distance (Iyer et al., 2021) as the information gain function results in the slection of a narrow sliver of data from the 2-D space (Fig. 2(d)). Moderate coresets (Xia et al., 2023) also sample from a narrow area in the distribution, resulting in poor diversity, but a slightly better balance between easy and difficult samples (Fig. 2(e)). Finally, with our proposed method, the diversity remains high and the distribution of difficulty scores in the coreset is also balanced (Fig. 2(f)).

# E  LIMITATIONS & LICENSE

## E.1  LIMITATIONS

**Access to Full Dataset & Pretrained Model.** Similar to the many previous coreset selection methods, our method relies on a model that has been pretrained or finetuned on the full dataset. We leverage the pretrained embeddings as well as the difficulty scores from this model. In doing so, we risk capturing the biases of the model. Further, one cannot use $\mathbb{D}^2$ PRUNING to create datasets from scratch and reduce annotation costs by avoiding redundant samples in the dataset. We note that an ideal data pruning method would not rely on access to the full dataset so that it can be used for creating challenging and effective datasets in a cost-effective manner. Our experiments in self-supervised and unsupervised data selection show promising results in this direction.

## E.2 LICENSE

We will publicly release our code and models. We use standard licenses from the community and provide the following links to the licenses for the datasets that we used in the project.

**CIFAR10, CIFAR100:** Other
**Adversarial NLI:** Creative Commons
**ImDB Reviews:** Other
**Counterfactual ImDB, NLI:** Apache
**DataComp:** MIT

