# OpenReview forum: "$\mathbb{D}^2$ Pruning: Message Passing for Balancing Diversity & Difficulty in Data Pruning"
_ICLR.cc/2024/Conference — ICLR 2024 poster_

### Official Review · Reviewer_3E93 · 2023-10-31

**Soundness:** 3 good
**Presentation:** 3 good
**Contribution:** 2 fair
**Rating:** 6
**Confidence:** 4

**Summary:**

This paper introduces a novel coreset selection algorithm called D2 pruning (Diversity-Difficulty pruning), which leverages undirected graphs and message passing to calculate difficulty score. The algorithm's primary goal is to tackle two important aspects: example difficulty and diversity within the selected subset of data points. D2 pruning works in the following way: 1. Graph Initialization: Nodes in graph G represent dataset examples and are connected to their k-closest neighbors in the embedding space. 2. Update difficulty score: Use message passing on the graph to update difficulty scores based on neighbor distance and difficulty.  3. Coreset Selection: Iteratively select balanced samples from high-density low-difficulty and low-density high-difficulty regions. Down-weight neighbors of selected samples to promote coreset diversity.

**Strengths:**

1. The authors proposed a novel coreset selection algorithm that aims to unify the benefits of data diversity and data difficulty. The proposed method is intuitive.

2. The proposed method is also evaluated on NLP datasets - lacked in prior work.

3. The evaluation compares $D^2$ with various baselines and shows that $D^2$ achieves better or comparable performance than SOTA methods.

4. The writing is good and easy to follow.

**Weaknesses:**

1. The performance improvement seems marginal. In most cases, the improvement is less than $1\%$.

2. With such performance differences, repeated evaluation can be suggested to mitigate the variance in the model training.

3. $D^2$ introduces some additional hyper-parameter, which may increase the coreset selection cost.

**Questions:**

1. How many iterations will the forward message passing phase have?

2. What is the importance score in fig 2 (what metrics used)?

3. It seems that $D^2$ can be combined with other metrics. Do you run an ablation study to see how $D^2$ performs with other importance scores?

---

> ### Author Response · Authors · 2023-11-18
> **Response to Review**
>
> Dear Reviewer 3E93,
>
> We thank you for your time, effort, and insightful comments. Please see our response below:
>
> * **Marginal improvements**: We see relatively small gains for easy datasets like CIFAR10 and redundant NLP datasets for the finetuning task because the performance of random selection on these datasets is already quite high. However, we see larger gains for difficult datasets like CIFAR100, ImageNet-1K, and DataComp. Importantly, we think that $\mathbf{D}^2$ Pruning will be a useful framework for future research into *self-supervised* and *unsupervised data selection* approaches, which are crucial topics in contemporary research for training foundation models. due to its plug-and-play nature, *where most existing work in coreset selection is no longer applicable*. It will benefit from any work that investigates better importance scores or meaningful representation embeddings. Moreover, it is not only flexible but also more scalable for large datasets than many coreset selection methods that rely on sub-modular functions, as we discuss in the General Response.
>
> * **Statistical analysis**: We have included a discussion of the significance of the scores in the revised pdf in **Sec. 5.1**. Our improvements on ImageNet-1K, DataComp, and CIFAR100 are significant (p<0.05, computed using a bootstrap of 100K samples [1, 2]). The improvement margins on CIFAR10 and NLP datasets are smaller due to random selection already doing so well on such datasets and the p-value is sometimes not significant for those improvements.
>
> * **Additional Hyperparameters**:  Coreset selection methods that do not operate solely on the ranking of an importance score need a hyperparameter, especially when combining the influence of two different functions, as we do in $\mathbf{D}^2$ Pruning. [3] combine facility location function for diversity with entropy score and use a weight parameter to balance them out. [4] perform a sweep over the number of bins and a cut-off ratio to remove bad examples. [5] perform bi-level optimization that occurs over multiple rounds of training and testing models. [6] tune the number of clusters used to get difficulty scores. It is definitely better to have a method without hyperparameters, but much further research is needed to get to that point.
>
> * **Iterations in message passing**: We use a single iteration of message passing in our main results. We conducted ablation experiments where ran multiple rounds of message passing and found that it does not benefit data selection due to aggressive smoothing of importance scores over the spatial dimension. Please see results in **Table 8 and Figure 5 in Appendix** of the revised pdf, and the General Response for a discussion.
>
> * **Importance Score in Fig. 2**: We use the forgetting score in Fig. 2, we have fixed the caption in the revised pdf to answer this question.
>
> * **Other importance scores**: Yes, we ran experiments with scores other than forgetting scores and we report those results in **Table 8 in the Appendix**. Please see General Response for the discussion.
>
> [1] Computer-intensive methods for testing hypotheses
>
> [2] An introduction to the bootstrap
>
> [3] Accelerating batch active learning using continual learning techniques.
>
> [4] Coverage-centric coreset selection for high pruning rates.
>
> [5] Glister: Generalization based data subset selection for efficient and robust learning.
>
> [6] Beyond neural scaling laws: beating power law scaling via data pruning.

---

> > ### Comment · Reviewer_3E93 · 2023-11-21
> > **Thanks for the response**
> >
> > Thank the authors for the further clarification and additional experiments. The clarification and additional evaluation address most of my concerns.
> >
> > Overall, I think the paper proposes a novel coreset selection, which explores how to jointly consider diversity and difficulty in coreset selection. The discussion on self-supervised/unsupervised coreset selection and NLP coreset selection also contributes to the community. However, I feel that the insignificant performance improvement can hurt the contribution of the paper, so I decided to keep my rating (6) unchanged.

---

### Official Review · Reviewer_niHR · 2023-11-01

**Soundness:** 3 good
**Presentation:** 3 good
**Contribution:** 2 fair
**Rating:** 6
**Confidence:** 3

**Summary:**

This paper proposes to balance both difficulty and diversity in the data sampling process. The authors argue that difficulty and diversity have been independently optimized but should be optimized together. To this end, this work proposes a graph-based method, D^2 pruning, which builds the graph based on data diversity and uses message passing to get information difficulty.

**Strengths:**

The authors perform extensive experiments, on both vision and NLP datasets, and compare to a well-covered set of baseline. Empirical results appear to be promising.  The methodology itself also appears to be interesting and novel, to the best of my knowledge.

**Weaknesses:**

The authors use distance in embedding space to construct the near-neighbor graph. However, there is a lack of support for why distance in embedding space is a good indicator of diversity. For example, what if we use another model to generate the embedding? What’s the influence between feature level embedding versus final layer embedding? Similarly, the authors use the forgetting score as the difficulty indicator. There is no discussion on why the author choose forgetting score? What would be the influence of choosing another score, such as a consistency score or loss of an approximate model?

**Questions:**

See weaknesses.

---

> ### Author Response · Authors · 2023-11-18
> **Response to Review**
>
> Dear Reviewer niHR,
>
>
> We thank you for your time, effort, and insightful comments. Please see our response below:
>
> * **Distance in embedding space as an indicator of diversity**:  Embeddings from pre-trained models have been widely used as representations of the semantic content of a sample within a high-dimensional sample for vision and text modalities. Distance or similarity computations between such embeddings have been successfully used for performing semantic similarity tasks like k-nearest neighbor selection [1], sentence similarity [2], etc. In our work, we refer to diversity in a data subset as the diversity of semantic content of the data samples in the subset. Hence, we use cosine similarity between embeddings as a mark of semantic similarity of two data samples and aim to select samples with maximum diversity based on this definition. We perform ablation experiments using different sources of embeddings and find that the embedding from the last layer of the ResNet18 (polling layer before classifier) works better than features from the deeper layer for vision models, and the [CLS] token embedding in RoBERTa works better than non-[CLS] token embeddings for NLP datasets. See general response and **Sec. D.2,  Table 8 in Appendix** in the revised pdf.
>
> * **Importance Score**: Similarly, we also perform ablation experiments for different importance scores, and find that the entropy score benefits selection from CIFAR10 while the EL2N score improves performance on the Adversarial NLI dataset. See general response.
>
> [1] Beating power law scaling via data pruning
>
> [2] Sentence-BERT: Sentence Embeddings using Siamese BERT-Networks

---

> > ### Author Response · Authors · 2023-11-22
> > **Follow-Up**
> >
> > Dear Reviewer,
> >
> > We would like to follow-up to see if our reply has adequately addressed your concerns or if you have any further questions. We are happy to provide additional clarifications if our response has not addressed your questions, kindly let us know. Thank you once again!

---

> > > ### Comment · Reviewer_niHR · 2023-11-23
> > > **Response to the authors**
> > >
> > > Thanks for the response. I appreciate the ablation experiments, which provide empirical evidence on some of the choices. I would increase the score.

---

### Official Review · Reviewer_2S5F · 2023-11-01

**Soundness:** 3 good
**Presentation:** 4 excellent
**Contribution:** 2 fair
**Rating:** 6
**Confidence:** 5

**Summary:**

This work proposes a new data pruning model called D^2 pruning that balances diversity and difficulty via a message-passing algorithm.
They show the performance superiority of the proposed method on vision and NLP datasets with supervised and self-supervised variants.

**Strengths:**

- Clear presentation and easy-to-follow writing.
- The algorithm is straightforward and easy-to-implement.
- The evaluation is extensive, with many datasets in multiple tasks, and solid.

**Weaknesses:**

- No time complexity analysis. The proposed algorithm based on message-passing seems to take quite a lot of time. The author should provide the time-complexity analysis with the exact GPU time taken because a data pruning method that takes too long time is less practical.
- No theoretical analysis. How this message-passing algorithm can guarantee better generalization than other baselines? Although the author provides some intuition (data pruning should consider both diversity and difficulty), why it should be achieved by the message-passing and how it can reduce the generalization error in Eq.(1) is missing.

**Questions:**

I think this work is also related but missed in discussion/comparison.

[a] Active learning is a strong baseline for data subset selection. NeurIPS workshop, 2022

---

> ### Author Response · Authors · 2023-11-18
> **Response to Review**
>
> Dear Reviewer 2S5F,
>
> We thank you for your time, effort, and insightful comments. Please see our response below:
> * **Time Complexity Analysis**: We have revised the pdf to include a time complexity analysis in **Sec. 4**, to complement the GPU time estimates that we had already provided in **Table 6 in the Appendix in original pdf**. D2 Pruning runs in mere minutes for CIFAR10, CIFAR100 datasets and takes ~23 minutes for ImageNet-1K when we use exact $k$-NN computation for initializing the graph. However, this time estimate is significantly improved when we use approximate $k$-NN using faiss to prune the DataComp dataset consisting of 12.8 million samples, i.e., $\mathbf{D}^2$ Pruning takes ~1 hour to prune the DataComp dataset. This time overhead is negligible compared to the training time of the models on such datasets and is a good investment if it yields improvements in downstream task performances. $\mathbf{D}^2$ Pruning is a scalable method that can be used for even larger datasets with modest memory requirements ($\mathcal{O}(nk)$). In comparison, most existing coreset selection methods are not scalable to large datasets or applicable to self-supervised approaches. Many contemporary self-supervised approaches for pruning datasets also involve similarly time-intensive preprocessing steps [1, 2, 3].
>
> * **Theoretical Analysis**: Our method design is motivated by the intuition that difficulty as well as diversity are needed to ensure the most representative data subset. While we do not have a theoretical analysis for this method, we provide a solid analysis of how this method works and why it is better than other methods through **Figures 2,3,5**, which provide insights for further research into the use of graphs for data subset selection.
>
> * **Additional reference**: Thank you for the reference, we have added this to the discussion in the related work section. The CCS method that we use as one of our baselines in the paper is as good or better than the method suggested in this work.
>
>
> [1] SemDeDup: Data-Efficient Learning at Web-scale through semantic deduplication
>
> [2] T-mars: Improving Visual Representations by Circumventing Text Feature Learning
>
> [3] Multimodal Dataset Pruning Using Image Captioning Models

---

> ### Comment · Reviewer_2S5F · 2023-11-20
> **Response to the rebuttal**
>
> Thanks to the author's comprehensive rebuttal. I've read all the responses, revised paper, and other reviewer's comments.
> Although I'm satisfied with the abundant content, I still have a few questions.
>
> **1. Time complexity**
>
> 1.1 **Why does the graph initialization take O(kn)?** Could you provide a more detailed analysis of this? As far as I know, the time complexity of the KNN algorithm for a single query point is O(nd), where n is the number of training examples and d is the number of features. Thus, to find the kNN of all data examples, it would take O(d*n^2). Similarly, In Table 6, 'Graph Creation' takes longer time than 'Iterative Selection', which also takes O(kn) according to authors, for all datasets. Please convince me if I misunderstood anything.
>
> 1.2 In Table 6, it would be better to **provide the training time** on the full dataset, and that with D^2 pruning for further clarification.
>
> **2. Theoretical analysis**
>
> 2.1 In my thought, only Figure 2 can be related to my question. D^2 pruning achieved better diversity and uncertainty than Facility Location, Graph Cut, and Moderate, while they are not a hybrid approach considering both diversity & uncertainty. Then, **why D^2 pruning is better than BADGE in terms of generalization?** An intuitive explanation or analysis for this would increase the novelty of this paper compared to the existing hybrid approaches like BADGE.
>
> Thanks!

---

> > ### Author Response · Authors · 2023-11-22
> > **Response**
> >
> > Thank you for taking the time to see our revisions and comments from other reviewers. We really appreciate your engagement in this rebuttal process and allowing us a chance to improve our paper! To answer your questions,
> >
> > **1) Time complexity**
> >
> > Thank you for identifying the mistake, the time complexity for graph initialization is erroneously mentioned as $\mathcal{O}(kn)$ in Sec. 4. It is indeed $\mathcal{O}(vn^2)$ where v is the vector dimension when the kNN is computed using the full dataset. We have revised the paper to reflect this change. Correspondingly, the time estimates in Table 6 for Graph initialization are higher than iterative selection because graph initialization is a more time-intensive step than iterative selection. The time estimates for graph creation of different datasets are not strictly comparable to each other because we run jobs of different batch sizes according to the size of the datasets to prevent OOM issues while computing the similarity matrix. We have also edited Table 6 to include training time for each dataset after data selection.
> >
> > **2) Novelty of $\mathbf{D}^2$ Pruning**
> >
> > Importance scores distributions tend to be skewed towards low scores, an example of the distribution of forgetting scores can be seen in Fig. 5(a). However, an easy sample surrounded by many other easy samples in the embedding space is less important than an easy sample that is surrounded by many difficult samples in the embedding space. This is because the area containing many easy samples is easy to learn for the model whereas the area containing difficult samples is hard to learn for the model. Intuitively, we want to include many samples from the hard-to-learn areas (even the easy samples) while sufficiently representing the easy-to-learn areas to prevent drops in accuracy for samples lying in that area [1]. Existing methods like BADGE and CAL-SDS2 [2] select the most important sample that is also representative of the complete set of samples; they do not make a distinction between easy-to-learn and hard-to-learn areas in the data representation space. Whereas, the forward message passing step in $\mathbf{D}^2$ Pruning increases the importance of a sample by an amount that is proportional to the importance scores of the samples surrounding it, thus ranking an easy sample in a hard-to-learn area higher than that in an easy-to-learn area. Effectively, the forward message passing step recalibrates the importance scores by taking the difficulty of the sample's local neighborhood into account and results in a less skewed distribution as we demonstrate in Fig. 5(b). The samples are ranked by this recalibrated importance score (represented by the updated node features in the graph) for the reverse message passing step.
> >
> > Additionally, in the reverse message passing step, after selection of each data sample, only its nearest neighbor samples are updated. This allows us to maintain a constant selection time for the same data subset size, with increasing size of the original dataset and makes $\mathbf{D}^2$ Pruning relatively more scalable for large-scale datasets, as we demonstrate with the DataComp dataset.
> >
> > To summarize, $\mathbf{D}^2$ Pruning is a scalable data selection algorithm that effectively re-ranks samples based on easy-to-learn and heard-to-learn areas in the embedding space via forward message passing and then imposes diversity in data selection via reverse message passing.
> >
> > [1] Beyond neural scaling laws: beating power law scaling via data pruning
> > [2] Accelerating Batch Active Learning Using Continual Learning Techniques (TMLR/DMLR@ICML'23)

---

> > > ### Comment · Reviewer_2S5F · 2023-11-23
> > > **Response to the rebuttal**
> > >
> > > Thanks for the additional response and the abundant rebuttal!
> > >
> > > Overall, I think the paper has **strength in empirical performance**, setting up state-of-the-art results in many benchmarks,  **applicability** to self-supervised/unsupervised coreset selection in vision and NLP tasks, and its **scalability** to large-scale datasets like ImageNet-1k and Datacomp. The paper also provides comprehensive ablation studies. Please include our discussion about the novelty in the final version. I decided to increase my rating to (6).
> > >
> > > Thanks.

---

### Official Review · Reviewer_4iU9 · 2023-11-04

**Soundness:** 3 good
**Presentation:** 3 good
**Contribution:** 3 good
**Rating:** 5
**Confidence:** 4

**Summary:**

This paper starts by emphasizing the significance of maintaining a balance between the diversity and difficulty of samples used in data subset selection, for speeding up the training process. To this end, the authors initially illustrate instances in which diversity sampling can result in an excessive over-sampling (attributable to over-representation) from regions characterized by relatively low complexity. Following this, as shown in Figure~2, the authors introduce a graph-based algorithm designed to select training examples that maintain a balance between diversity and difficulty. The algorithm is based on message parsing on the constructed graph where each datapoint is a node. Subsequently, the paper provides experimental evidence in support of their proposed method, conducted across a range of datasets encompassing both vision and language domains, as well as their joint modality. The experimentation includes scenarios involving both supervised and self-supervised learning. The paper also addresses the interesting datacomp setting and evaluates its performance against the corresponding benchmarks, including VTAB and retrieval.

**Strengths:**

A wide range of data modalities is considered which I particularly appreciate, unlike other papers in the community. Ablation is also done on the hyperparameters mentioned in the proposed algorithm, which include $\gamma_r$, $\gamma_f$ (kernel width), and the sparsity of the graph. The paper was pretty much straightforward and they mentioned that $D^2$ provides boosts under a low to medium "pruning" regime. Overall, I like the simplicity of writing.

**Weaknesses:**

I feel that the prime weakness of this work is from the angle of related works, and baselines. Highlighting the importance of balancing diversity and difficulty is not new, and indeed if one is using not well-tuned diversity selection methods, then they won't be able to give a full representation of the dataset (minor modes which are not outliers but are difficult). That being said, I need an explanation, and if possible, results on the following baselines --

For general supervised cases

- CRAIG [1]
- GLISTER (this paper has the same motivation as mentioned in Eq1) [2]
- GradMatch [3]
- Top-k method [4]: Another graph-based sampling that downweighs the contribution based on neighbors.
- CREST (an extension of CRAIG) [5]

A combination of submodularity with difficulty has been explored in the following --

- FASS (two-stage procedure for active learning, but a similar two-stage procedure can be considered here) [6]
- A combination of submodularity and difficulty has been considered in CAL-SDS2 [7]
- MCL (Combination of hardness and diversity for curriculum learning) [8]
- DIHCL (another combination of hardness and diversity for curriculum learning) [9]

More recent works on diversity-based selection (for NLP) --
- MILO [10]
- INGENIOUS [11]

On SSL:
- See SAS [12]

On multimodality see T-MARS [13]

Concluding thoughts:
I believe the paper should discuss these works and should include some of them as baseline.

References
- [1] Data-efficient Training of Machine Learning Models (ICML'20)
- [2] GLISTER: Generalization-based Data Subset Selection for Efficient and Robust Learning (AAAI'21)
- [3] GRAD-MATCH: Gradient Matching based Data Subset Selection for Efficient Deep Model Training (ICML'21)
- [4] SELECTIVE ANNOTATION MAKES LANGUAGE MODELS BETTER FEW-SHOT LEARNERS (ICLR'23)
- [5] Towards Sustainable Learning: Coresets for Data-efficient Deep Learning (ICML'23)
- [6] Submodularity in Data Subset Selection and Active Learning (ICML'15)
- [7] Accelerating Batch Active Learning Using Continual Learning Techniques (TMLR/DMLR@ICML'23)
- [8] Minimax Curriculum Learning: Machine Teaching with Desirable Difficulties and Scheduled Diversity (ICLR'18)
- [9] Curriculum Learning by Dynamic Instance Hardness (NeurIPS'19)
- [10] MILO: Model-Agnostic Subset Selection Framework for Efficient Model Training and Tuning
- [11] INGENIOUS: Using Informative Data Subsets for Efficient Pre-Training of Language Models
- [12] Data-Efficient Contrastive Self-supervised Learning: Most Beneficial Examples for Supervised Learning Contribute the Least (ICML'23)
-  [13] T-MARS: Improving Visual Representations by Circumventing Text Feature Learning

**Questions:**

- For the difficulty-based methods, in supervised settings, methods such as forgetting-event are clear that they use the learning dynamics. However, methods such as -- entropy, EL2N (which is similar to the norm of the gradient concerning bias term), and area under margin score, can be computed during any step of training. Therefore, does the paper consider the moving average of the dynamics (of some pre-trained models, for which they've assumed the access), or is it taken at the end of the training? In case it is moving average, I am okay with it, but if it is taken at the end of the training, then the training set methods like entropy/margin/EL2N can be very wrong in judging the hardness.


- Can authors provide standard deviation or statistical analysis in cases where the second-best technique is very close?
- For BADGE does the author use true labels instead of pseudo labels? BADGE was proposed in Active learning and hence it is important to make sure it doesn't have a disadvantage in supervised setting comparisons.
- k and s_k are overloaded as expressions in the paragraph above equation 6.
- What happens when one runs message parsing for more than one round? Can authors provide an experiment on that or justification?
- Eq. 1 theta should rather me $\theta^*(S')$ to show that it is a solution to the optimization.

---

> ### Author Response · Authors · 2023-11-18
> **Response to Review**
>
> Dear Reviewer 4iU9,
>
> We thank you for your time, effort, and insightful comments. Please see our response below:
>
> * **Baselines**: We would like to thank the reviewer for putting together a detailed list of the works relevant to ours and outlining the methods for our convenience. We agree that these methods are relevant to our work. So, we picked four additional baselines for experiments (see general response).
>
> * **Discussion on suggested baselines**: We have added discussion on new baselines and other suggested references in **Sec. 5.1, Sec. 5.3 and Sec. 6**, as much as possible in the limited space. Additionally, we have updated Figure 2 to show results from data selection using diversity-only approaches using facility location (FL) and graph-cut (GC) sub-modular functions [1]. As we discuss in Sec. 5.1, the main drawback of using FL or GC submodular functions in any form is the need for access to the complete similarity matrix that causes a memory requirement of $\mathcal{O}(n^2)$ [2], which is not scalable for larger datasets.
>
> * **Moving Averages**: Yes, we consider the moving average of El2N, AUM scores, etc.
> * **Statistical analysis**: We have included a discussion of the significance of the scores in the revised pdf in **Sec. 5.1**. Our improvements on ImageNet-1K, DataComp, and CIFAR100 are significant (p<0.05, computed using bootstrap of 100K samples [3, 4]). The improvement margins on CIFAR10 and NLP datasets are smaller due to random selection already doing so well on such datasets and the p-value is sometimes not significant for those improvements.
> * **BADGE**: We use the true labels in our implementation of BADGE to make a fair comparison.
> * **$x_k$, $s_k$**: Thank you for catching this error, we have fixed this in the revised version.
> * **Message passing for more than one round**: Please see general response.
> * **Eq 1**. Theta: Thank you for pointing this out, we have fixed this in the revised pdf.
>
>
> [1] Submodular combinatorial information measures with applications in machine learning
>
> [2] https://apricot-select.readthedocs.io/en/latest/index.html
>
> [3] Computer-intensive methods for testing hypotheses
>
> [4] An introduction to the bootstrap

---

> ### Comment · Reviewer_4iU9 · 2023-11-19
> **Thanks for adding the baselines**
>
> I have a few questions:
>
> 1. For the Facility location function it is possible to use sparse matrices (in fact as long as entries are non-negative function is submodular). Therefore, did the experiments use the sparse kernels (the sparse similarity matrix that was used in the graph for D^2)? Therefore, I think it is not okay to write FL always needs $\mathcal{O}(n^2)$ space.
>
> 2. CAL-SDS2 can be used with any hardness metric, and in general - $F(A) = FL(A) + \lambda \operatorname{log}(1 + m(A))$ where $m(A)$ is a function for hardness --  entropy, margin, EL2N, etc. So I still feel that additional exploration on that front can be done.

---

> > ### Author Response · Authors · 2023-11-21
> > **Response to questions**
> >
> > Thank you for taking the time to respond to our rebuttal! We appreciate your engagement in this rebuttal process and allowing us a chance to improve our paper. To answer your questions,
> >
> >
> > 1. Thank you for clarifying the possibility of using sparse matrices in facility location functions for us. We used the full similarity matrix for Facility location, but we are re-running our experiments using the sparse similarity matrix. We implemented our version of CAL-SDS2 using pre-defined functions and optimizers in [1]; if the reviewer is aware of a codebase that directly implements CAL-SDS2 using sparse matrices, we request them to kindly point us to it.
> >
> > 2. We have launched jobs for tuning the hyperparameters of CAL-SDS2 [$\sigma$, $\lambda$] under different importance scores and nearest-neighbor similarity matrices as used for $\mathbf{D}^2$ Pruning, and so far we haven't seen better results than what we have reported using entropy + full similarity matrix. We will report the results again as soon as our experiments are completed. We would also like to mention that we did not come across CAL-SDS2 in our literature review for coreset selection methods that balance difficulty as well as diversity because the paper focuses on active/continual learning. We wish we had sufficient time to explore CAL-SDS2, however, the rebuttal period is not sufficient to conduct *fair* comparative experiments. We agree with the reviewer that submodular functions can be used to balance difficulty and diversity, and we will amply cover this in our experiments and discussion section. However, the goal of our work is to show that representing datasets as graphs *also* works as an effective framework for balancing difficulty + diversity in data selection, not just in the scenario of one-shot coreset selection but in self-supervised data pruning and unsupervised data filtering as well. It is an interesting line of research because it opens up possibilities for using graph neural networks (GNNs) to not just prune but also understand datasets, and our work is a proof of concept in this direction. Therefore, we request the reviewer to consider our work independently of the work on sub-modular functions.
> >
> > [1] https://apricot-select.readthedocs.io/en/latest/index.html

---

> > > ### Comment · Reviewer_4iU9 · 2023-11-21
> > >
> > > 1. I unfortunately don't know any implementation that takes care of a sparse (CSR/COO) matrix for the mixture of FL and hardness. That being said [1] does offer support for sparse matrices. When I meant sparsification using k-nn like the proposed graph-based approach, one can emulate the effect of a sparse matrix by having float 0s in a dense (n,n) matrix. I would like the authors to clarify in the paper that FL can be used with sparse matrix, and it is not correct to say that it always uses $\mathcal{O}(n^2)$ space. (The reason I am mentioning this is because in the paper the way it is written implies that FL cannot be done in less than quadratic memory)
> > >
> > > 2. I agree that CAL was proposed in the AL regime. However, AL has been used in the past for data subset selection. That said, I strongly recommend performing a search on hyperparameters of the baseline, as I feel that the current mentioned results may be unfair (as entropy might not be the right metric). Methods like submodularity + hardness are very intuitive compared to the proposed technique.
> > >
> > > Additional Comment:
> > >
> > > I don't see any discussion other than one line in related works for [2]. I recommend adding a detailed discussion for this graph-based technique, as this work is quite similar, although done in in-context learning (which doesn't mean it cannot be used for regular data subset selection).
> > >
> > > [1] https://submodlib.readthedocs.io/en/latest/functions/facilityLocation.html
> > > [2] SELECTIVE ANNOTATION MAKES LANGUAGE MODELS BETTER FEW-SHOT LEARNERS

---

> ### Author Response · Authors · 2023-11-22
> **Response**
>
> Thank you for the follow-up comments!
>
> 1) Thanks for the suggestion, we are running experiments using the sparse similarity matrix where only k-nearest neighbor distances are non-zero and everything else is 0. We have also revised our paper to remove the lines that say that FL always needs $\mathcal{O(n^2)}$ space. We have also expanded our discussion of the Top-$k$ Voting method in the Related Work section. Top-$k$ voting is similar to the reverse message passing step in our method and only ensures diversity in selection. $\mathbf{D}^2$ Pruning first combines the influence of importance scores and local neighborhood in the forward message passing step and then imposes diversity in the reverse message passing step.
>
>
> 2) We also agree that CAL-SDS2 warrants a proper search for hyper-parameters as well as experiments with various importance scores to get the best result. In our current experiments, we are performing a grid search over $\lambda$ = $\[1, 2, ... 10]$ and $\sigma$ = $[0.1, 0.2, 0.3, ... 1.0]$ for entropy, forgetting and EL2N scores. The best results so far for CIFAR10 are as follows:
>
> | Method | 30% | 50% | 70% | 80% | 90% |
> | ----------- | ----------- | ----------- |  ----------- |  ----------- |  ----------- |
> | Random |   94.3  | 93.4 | 90.9 | 88.0 | 79.0 |
> | $\mathbf{D}^2$ Pruning  |  95.7 | 94.9 | 93.3 | 91.4 | 87.1 |
> | CAL-SDS2 [Entropy] |  95.0 | 93.8 | 91.6 | 87.1 | 81.5 |
> | CAL-SDS2 [Forgetting]  |  95.7 | 94.4 | 92.1 | 88.9 | 84.6 |
> | CAL-SDS2 [EL2N]  |  94.8 | 94.1 | 92.2 | 87.8 | 83.5 |
>
> We see improved performance from CAL-SDS2 with Forgetting score at all pruning rates and CAL-SDS2 performs quite well as low pruning rates, but it still struggles at high pruning rates, even with Forgetting score as the difficulty metric. We will continue tuning our results with CAL-SDS2 and definitely add the improved results to our paper when the experiments end.
>
> We would like to reiterate the intuition behind $\mathbf{D}^2$ Pruning:
>
> Importance score distributions tend to be skewed towards low scores, an example of the distribution of forgetting scores can be seen in Fig. 5(a) in the Appendix. However, an easy sample surrounded by many other easy samples in the embedding space is less important than an easy sample that is surrounded by many difficult samples in the embedding space. This is because the area containing many easy samples is easy to learn for the model whereas the area containing difficult samples is hard to learn for the model. Intuitively, we want to include many samples from the hard-to-learn areas (even the easy samples) while sufficiently representing the easy-to-learn areas to prevent drops in accuracy for samples lying in that area [1]. Existing methods like BADGE and CAL-SDS2 select the most important sample that is also representative of the complete set of samples; they do not make a distinction between easy-to-learn and hard-to-learn areas in the data representation space. Whereas, the forward message passing step in $\mathbf{D}^2$ Pruning increases the importance of a sample by an amount that is proportional to the importance scores of the samples surrounding it, thus ranking an easy sample in a hard-to-learn area higher than that in an easy-to-learn area. Effectively, the forward message passing step recalibrates the importance scores by taking the difficulty of the sample's local neighborhood into account and results in a less skewed distribution as we demonstrate in Fig. 5(b). The samples are ranked by this recalibrated importance score (represented by the updated node features in the graph) for the reverse message passing step.
>
> Additionally, in the reverse message passing step, after the selection of each data sample, only its nearest neighbor samples are updated. This allows us to maintain a constant selection time for the same data subset size, with increasing size of the original dataset, and makes $\mathbf{D}^2$ Pruning relatively more scalable for large-scale datasets, as we demonstrate with the DataComp dataset.
>
> To summarize, $\mathbf{D}^2$ Pruning is a scalable data selection algorithm that effectively re-ranks samples based on easy-to-learn and hard-to-learn areas in the embedding space via forward message passing and then imposes diversity in data selection via reverse message passing.

---

> > ### Comment · Reviewer_4iU9 · 2023-11-23
> > **Quick Clarification**
> >
> > Thanks for providing the additional results!
> >
> > In the results above, does the FL in CAL-SDS2 use a sparse matrix (where sparsity is just zeroed-out entries and not a CSR/COO matrix)??

---

> > > ### Author Response · Authors · 2023-11-23
> > >
> > > Thanks for your reply! No, in these experiments, we are using the full matrix. We do have another set of experiments for the use of sparse matrix in FL which is underway and will take loner for tuning, unfortunately.

---

> > > > ### Comment · Reviewer_4iU9 · 2023-11-23
> > > > **Increasing the rating.**
> > > >
> > > > Thank you for the response and additional results. I do hope that the authors include the results with sparse FL-based CAL-SDS2, and in the final paper include a discussion on ablations with different hardness metrics. That being said, I am increasing my rating but won't be able to increase it any further, given that sparse matrix-based indeed are important.
> > > >
> > > >
> > > > Thanks!

---

### Official Review · Reviewer_DbMJ · 2023-11-10

**Soundness:** 3 good
**Presentation:** 2 fair
**Contribution:** 2 fair
**Rating:** 5
**Confidence:** 5

**Summary:**

The $\mathbb{D}^2$ PRUNING method presented in this paper demonstrates a novel approach for selecting the most useful data from large training sets (coresets) for deep learning model training.

It combines two key factors: data diversity and sample difficulty. The method represents the training dataset as a graph and uses a message-passing algorithm to update each data point's difficulty score by considering its neighbors. This process ensures a balance of diverse and challenging data in the selected coreset.

$\mathbb{D}^2$ PRUNING has shown to be effective in improving model performance, particularly for image classification and natural language processing tasks, and is especially useful at low-to-medium data pruning rates.

**Strengths:**

1. The paper presents a novel approach - $\mathbb{D}^2$-PRUNING, which balances both data diversity and difficulty, along with its applicability to both supervised and self-supervised learning contexts. This positions it as a valuable tool in the ongoing evolution of corset selection techniques for data efficient learning.
2. The experiments are clear and the authors provide ablation experiments to support the theory and newly introduced hyper-parameters in the paper.
3. It is commendable that the paper divulges into the NLP domain and demonstrates improved performance over existing methods in coreset selection.

**Weaknesses:**

1. The paper demonstrates very incremental gains in performance over State-of-the-Art method (like Ash et al., 2019).
2. The experiments in section 5.1 do not compare $\mathbb{D}^2$-PRUNING State-of-the-Art methods discussed in (Guo et al., 2021) such as GLISTER (Killamsetty et al., 2021), CRAIG (Mirzasoleiman et al., 2020), GRAD-MATCH (Killamsetty et al., 2021) etc.

**Questions:**

1. The paper uses inconsistent numberings (A and then 2, 3) in section 1 which should be rectified.
2. An important investigation aspect for this paper would be to demonstrate performance on very small values of selection ratios $k$ as discussed in (Guo et al, 2021) and perform more than 1 message passing (K-shot setting).
3. The paper refers to additional information in the appendix section. It would greatly improve the readability of the paper if the authors point to exact section numbers in the appendix.
4. It is unclear if the parameter $\gamma$ mentioned in section 5.1 refers to $\gamma_f$ or $\gamma_r$.
5. Although optional, including an algorithmic view of the proposed approach would be interesting to clarify how $\mathbb{D}^2$-PRUNING fits into the training and evaluation process of deep-learning models.

---

> ### Author Response · Authors · 2023-11-18
> **Response to Review**
>
> Dear Reviewer DbMJ,
>
> We thank you for your time, effort, and insightful comments. Please our response below:
> * **Additional baselines**: Please see general response.
> * **K-shot setting**: Please see general response, under *Ablation Experiments*.
> * **Very small selection ratios**: Thank you for the suggestion, we ran experiments along these lines and report our results in **Table 7, Sec. D.1 in the Appendix**. We see large improvements using $\mathbf{D}^2$ Pruning in some cases, such as 3% over CCS at 99.5% pruning of CIFAR10. However, the improvements are not consistent, either from CCS or from our method, suggesting that diversity isn't the important factor at extremely low data budgets.
> * **Inconsistent numberings**: Thank you for catching this error, we have fixed it in the revised version.
> * **References to sections in Appendix**: Thank you for the suggestion, we have fixed this in the revised version.
> * **Section 5.1 Parameter**: The parameter $\gamma$ refers to $\gamma_{r}$. We have fixed this in the revised version.
> * **Algorithm**: Thank you for the suggestion, we have included a pseudo-code of $\mathbf{D}^2$ Pruning for data selection in **Algorithm 1 in the Appendix**. Please let us know if it doesn't answer your question correctly, we will be happy to provide a different answer according to your suggestion.
> * **Incremental gains**: We see relatively small gains for easy datasets like CIFAR10 and redundant NLP datasets for the finetuning task because the performance of random selection on these datasets is already quite high. However, we see larger gains for difficult datasets like CIFAR100, ImageNet-1K, and DataComp. Importantly, we think that D2 Pruning will be a useful framework for future research into *self-supervised* and *unsupervised data selection* approaches, which are crucial topics in contemporary research for training foundation models. due to its plug-and-play nature, *where most existing work in coreset selection is no longer applicable*. It will benefit from any work that investigates better importance scores or meaningful representation embeddings. Moreover, it is not only flexible but also more scalable for large datasets than many coreset selection methods that rely on sub-modular functions, as we discuss in the General Response.

---

> > ### Comment · Reviewer_DbMJ · 2023-11-22
> >
> > I would like to thank the authors for addressing some of the questions and weaknesses in the paper.
> > Unfortunately, the provided clarifications still do not address my concerns about incremental gains. The observed results that, $\mathbb{D}^2$-Pruning produces "larger gains on difficult datasets" is not backed by theoretical evidence in the method nor in previous work. Thus I have decided to keep my rating (5) unchanged.

---

> > ### Author Response · Authors · 2023-11-22
> > **Follow-Up**
> >
> > Dear Reviewer,
> >
> > We would like to follow-up to see if our reply has adequately addressed your concerns or if you have any further questions. We are happy to provide additional clarifications if our response has not addressed your questions, kindly let us know. Thank you once again!

---

### Author Response · Authors · 2023-11-18
**General Response to Reviewers**

Dear reviewers,

We thank you for your time, effort, and insightful comments. The feedback has improved the experiments and analysis in our paper (see revised pdf; new text is in blue). Here, we address concerns raised by more than one reviewer and point to sections in the revised PDF wherever applicable. For other concerns, please see individual responses.

* **Results on Additional Baselines**: We have added four new baselines as per recommendations. See results in **Tables 1,2,3** and discussion in **Sec. 5.1, 5.3** in revised pdf:
  - GLISTER [6]: A bilevel-optimization approach for selecting robust coresets, which has been shown to outperform GRAD-MATCH, CRAIG, etc. in [1]
  - CAL-SDS2 [4]: Submodular optimization using a combination of entropy function and facility location [2] function for maximizing diversity; implemented using [3]
  - INGENIOUS [7]: Diversity-only approach via submodular optimization using the facility location function for NLP datasets
  - T-MARS [8]: A heuristic-based filtering approach for multimodal datasets where images containing text, which is also a major part of the caption, are filtered out.

  **Summary of Results**:
    - GLISTER performs as well as $\mathbf{D}^2$ Pruning for low pruning rates for vision datasets, but falters at medium, higher pruning rates.
    - CAL-SDS2 requires tuning of a hyperparameter to balance the effects of difficulty and diversity. We ran ten rounds of tuning for each run and found that it does not perform as well as $\mathbf{D}^2$ Pruning.
    - INGENIOUS merely maximizes diversity without considering the difficulty of the samples, and hence, achieves poor results for finetuning of RoBERTa on NLP datasets.
    - The facility location submodular function requires the full similarity matrix over the entire dataset with a $\mathcal{O}(n^2)$ memory requirement for optimization, which is simply not scalable for large datasets. In contrast, $\mathbf{D}^2$ Pruning can work with approximate $k$-NN for large datasets that does not require the full similarity matrix and only requires $\mathcal{O}(nk)$ memory during message passing.
    - T-MARS is a filtering technique that shows significant gains over CLIP-Score filtering of DataComp small. However, it is orthogonal to our goal of increasing diversity in the selected samples in $\mathbf{D}^2$ Pruning, $\mathbf{D}^2$ Pruning also yields large gains on top of T-MARS when applied to T-MARS filtered data, showing that our method is generally complementary to heuristics-based filtering approaches such as T-MARS, SIEVE [9], etc.

| Method | Avg. Accuracy on DataComp |
| ------- | ------ |
| No filtering | 13.2 |
| CLIP Score filtering | 16.0 (+2.8) |
| D2 Pruning | 17.0 (+3.8) |
| T-MARS [8] | 17.7 (+4.5) |
| T-MARS + D2 Pruning | 18.8 (+5.6) |

* **Ablation Experiments**:
  See results in **Table 8 in Appendix.**
  - **Multiple message passing iterations**: To better understand the effect of the forward message passing in $\mathbf{D}^2$ Pruning, we visualize a random subset of CIFAR10 samples in a 2-dimensional t-SNE embedding space in **Fig. 5 in Appendix** before and after forward message passing under various scenarios. See discussion in **Sec. D.2 in Appendix**. Multiple iterations of message passing at the same number of nearest neighbors ($k$) has an effect similar to that of Gaussian smoothing in the embedding
space because all nodes receive similar updates. As a result, the distribution of updated node features in $\mathbf{D}^2$ Pruning is biased towards a narrow spectrum of values that no longer benefits the data selection task.
  - **Importance Score**: We find that the entropy score benefits performance on CIFAR10 at higher pruning rates, whereas EL2N benefits performance on Adversarial NLI for low pruning rates. Importantly, we do not see large drops in performance with any of these score functions, suggesting that improved difficulty metrics can be paired with $\mathbf{D}^2$ Pruning for larger improvements.
  - **Embeddings**: We experiment with features from the last convolutional layer in ResNet18 for CIFAR10 and use the average of non-[CLS] tokens in RoBERTa for ImDB dataset, and find that neither source is as effective as the features extracted from the last layer of the pretrained model.

[1] Deepcore: A comprehensive library for coreset selection in deep learning.

[2] Submodular combinatorial information measures with applications in machine learning

[3] https://apricot-select.readthedocs.io/en/latest/index.html

[4] Accelerating Batch Active Learning Using Continual Learning Techniques

[5] Curriculum Learning by Dynamic Instance Hardness

[6] Glister: Generalization based data subset selection for efficient and robust learning

[7] INGENIOUS: Using Informative Data Subsets for Efficient Pre-Training of Large Language Models

[8] T-mars: Improving visual representations by circumventing text feature learning

[9] SIEVE: Multimodal Dataset Pruning Using Image Captioning Models

---

### Meta-Review · Area_Chair_kQPD · 2023-12-15

**Metareview:**

The abstract discusses a novel approach to coreset selection, which aims to optimize the training of deep learning models by selecting the most effective subset of training data. This method addresses the limitations of traditional approaches that focus either on data diversity or difficulty scoring. The authors represent the dataset as an undirected graph and employs a message-passing algorithm to update the difficulty scores of data samples, considering the difficulty of neighboring samples. This approach ensures that the selected coreset includes both diverse and challenging examples, leading to more efficient and effective model training. The method has been tested on various vision and NLP datasets, showing improved performance over existing methods, particularly at lower pruning rates. Additionally, Pruning enhances the diversity and generalization ability of models trained on large multimodal datasets, demonstrating its versatility and effectiveness in dataset processing and understanding.

The reviewers have pointed out several issues with the paper including, missing comparison to several important baselines (the authors do add some results in the rebuttals), lack of understanding/intuition on choice of diversity functions, and why some others can't be used. I would encourage the authors to take the suggestions into account while making the next version!

**Justification For Why Not Higher Score:**

Several issues raised by the reviewers should be addressed by the authors!

**Justification For Why Not Lower Score:**

N/A

---

### Decision · Program_Chairs · 2024-01-16

Accept (poster)